# Resonance energy transfer sensitises and monitors in situ switching of LOV2-based optogenetic actuators

Li-Li Li [1,2,4], Florence M. Klein[1], Lorenzo Li Greci [1], Arkadiusz Popinigis[1,5], Florian Freudenberg [3] & Michael J. Courtney [1,2✉]

Engineered light-dependent switches provide uniquely powerful opportunities to investigate and control cell regulatory mechanisms. Existing tools offer high spatiotemporal resolution, reversibility and repeatability. Cellular optogenetics applications remain limited with diffusible targets as the response of the actuator is difficult to independently validate. Blue light levels commonly needed for actuation can be cytotoxic, precluding long-term experiments. We describe a simple approach overcoming these obstacles. Resonance energy transfer can be used to constitutively or dynamically modulate actuation sensitivity. This simultaneously offers on-line monitoring of light-dependent switching and precise quantification of activation-relaxation properties in intact living cells. Applying this approach to different LOV2-based switches reveals that flanking sequences can lead to relaxation times up to 11-fold faster than anticipated. In situ–measured parameter values guide the design of target-inhibiting actuation trains with minimal blue-light exposure, and context-based optimisation can increase sensitivity and experimental throughput a further 10-fold without loss of temporal precision.

[1] Neuronal Signalling Lab, Turku Bioscience Centre, University of Turku and Åbo Academy University, Biocity, Turku, Finland. [2] Turku Screening Unit, Biocity, Turku, Finland. [3] Department of Psychiatry, Psychosomatic Medicine and Psychotherapy, University Hospital, Goethe University, Frankfurt, Germany. [4] Present address: Metabolic Research Laboratories, Wellcome-MRC Institute of Metabolic Science, University of Cambridge, Cambridge, UK. [5] Present address: BLIRT S.A., Trzy Lipy 3/1.38, 80-172 Gdansk, Poland. ✉email: michael.courtney@bioscience.fi

Optogenetic methods are increasingly important throughout biology, dissecting spatiotemporal properties of signalling networks and addressing fundamental questions in pathology, physiology and even psychology in intact organisms. Blue-light photoreceptors such as LOV2 are among the most popular cellular optogenetics photosensors[1]. However, blue light can perturb cells, potentially complicating long-term (hours-days) optogenetic modulation[1,2]. LOV2 can be indirectly sensitised by tuning relaxation[3], but this compromises time-resolution. Optogenetics has a unique potential to impose or modulate complex time-encoded inputs on cellular targets, and establish long-term manipulation and mimicry of periodic and pulsatile biological phenomena[4,5]. It would be valuable to increase sensitivity while maintaining time-resolution. A second difficulty optimising optogenetic regulators for new targets is that real-time in situ detection of photosensor switching has not been reported. Except in specific cases where protein translocation or other visualised downstream phenomena are targeted, it is hard to know whether an actuator has been switched. Relaxation rate constants are obtained by cell-free spectroscopy[3,6] and occasionally with heavily modified constructs not subsequently used for experimental actuation[7]. LOV2 designs that modify Jα, the moiety regulated by light-dependent switching, can accelerate cell-free relaxation >30-fold[8]. Rational design of illumination paradigms requires in situ relaxation and activation rates. Without this information, validation and use of optogenetic tools remain challenging tasks.

Here we present a simple method addressing these needs, simultaneously increasing light sensitivity of an optogenetic switch while providing on-line detection of activation and relaxation. Engineered resonance energy transfer (RET) between a mTurquoise2 fluorescent protein and FMN-bound AsLOV2 increases switch sensitivity. We demonstrate this with an optically-regulated nuclear export sequence (NES). Similar results are obtained with the optogenetic JNK inhibitor, optoJNKi. RET also causes mTurquoise2 quenching by dark-state AsLOV2. Blue-light generates the cysteine adduct that drives the actuator's conformational change, leading to dequenching. LOV2 switching is thus directly observed as fluorescent tag dequenching. Darkness allows cysteine adduct decay and recovery of ground-state AsLOV2, regenerating tag quenching. Dequench-requench dynamics reveals activation and relaxation rates of AsLOV2 switching in intact living cells. This generalises to other fluorescent protein tags and AsLOV2-based actuators. Strikingly, LOV2 relaxation in actuators was up to 11-fold faster than expected from AsLOV2 spectroscopy. The differences in relaxation time could be explained by the impact of sequences flanking the LOV2 domain rather than an effect of the in vivo environment. Selection of fluorescent tags and consideration of activation-relaxation parameters facilitates minimal photon exposure protocols for JNK pathway regulation. Finally, comparison of LOV2 FRET-derived parameters with target pathway kinetics guides optogenetic actuator optimisation to further reduce blue light requirements tenfold and increase experimental throughput tenfold. This facilitates repeatable and relatively non-invasive interrogation of bidirectional pathway dynamics that could not be achieved by pharmacological approaches. We anticipate the methodology described could promote wider, more efficient application of cellular optogenetics in diverse experimental settings.

## Results

**Fused fluorescent protein determines actuation sensitivity.**
Blue-light activated optogenetic tools are commonly tagged with red fluorescent proteins[6,9], verifying construct expression without activation. This occupies blue and yellow-green excitation channels, and multiplexing with fluorescent probes becomes difficult.

A cyan tag allows recording of actuator expression without occupying other channels, permitting multiplexing with yellow to infra-red channels. However, there might be unintended consequences—spectral overlap with proximal (within 5 nm) photosensor chromophores (flavins in LOV2 and cryptochromes) may permit RET, influencing actuator sensitivity. Comparison of spectra from LOV2 and cyan, yellow and red proteins suggests potential RET in all cases (Supplementary Fig. 1). Consequences could include sensitised actuation by cyan proteins, suppressed actuation by yellow and red proteins or no effect (Supplementary Fig. 2).

Determining the influence of fusion tags requires an assay to quantify LOV2 switching. The optogenetic nuclear translocator LEXY interweaves a NES with the AsLOV2-Jα helix[9], providing a convenient actuation readout. Light-induced unfolding of the Jα exposes the NES, increasing nuclear export until LOV2 relaxation terminates NES exposure[9]. We generated a panel of such constructs using different fluorescent tags (XFPs, Supplementary Figs. 3–4). Because of a minor sequence difference, we refer to these as optoNES to avoid possible confusion with the original LEXY[9]. We only consider light sensitivity rather than dynamic range of translocation; these are independent parameters just as enzyme $K_m$ and $V_{max}$ are independent. Hippocampal neurons infected with AAVs encoding XFP-optoNES (Supplementary Fig. 4) show localisation to nuclear and cytoplasmic compartments, but translocation out of the nucleus in response to light (Fig. 1a, b; Supplementary Fig. 5 and Supplementary Movies 1–6).

To quantify light sensitivity, we provided cycles of ten light pulses, each partially switching the LOV2 domain. Translocation rate is quantifiable by fluorescence imaging (Fig. 1a–d). Four cycles were applied at each of four 438 nm photon doses, between ~14 and 420 μmol m$^{-2}$ flash$^{-1}$ (1 μmol m$^{-2}$ s$^{-1}$ = 27 μW cm$^{-2}$; Fig. 1c). Sensitivity was calculated from the nucleocytoplasmic ratio change per flash for each dose (Fig. 1e and Supplementary Fig. 6). Under these conditions, all constructs exhibited light-dependent nuclear export. But the mTurquoise2 fusion was notably more sensitive to blue light than the other fusion constructs (Supplementary Table 1).

To determine whether these differences result from RET, we separated fluorophores and LOV2 domain with 24 residues (Supplementary Fig. 7), increasing inter-dipole distances to which RET efficiency is exquisitely sensitive (6th power relation, Supplementary Fig. 1A). Excess positive charge in the spacer should favour an extended confirmation, but even a random-coil conformation (with residue spacing of 0.38 nm[10]) would add 1.9 nm. Generously estimating initial inter-dipole distance at 5 nm, a typical value for $R_0$, this equates to 1.4-fold distancing and a minimum sevenfold reduction of RET ($[R/R_0]^6$).

These reduced-RET optoNES variants export with indistinguishable sensitivity (Fig. 1f and Supplementary Fig. 8). Thus, the spacer increased sensitivity where LOV2 can act as a donor (approximately threefold for Ypet or mScarlet constructs) and reduced sensitivity of the mTurquoise2 construct >twofold (Fig. 1e–h and Supplementary Table 1). To evaluate whether differences in fusion protein sequences, and not RET, could be responsible for sensitivity differences, we replaced the constitutively fluorescent proteins with reversibly photoswitchable protein Dronpa. This protein can be switched between 400 nm absorbing and yellow fluorescent protein-like states[11] (Supplementary Fig. 9A). Thus we compared actuation sensitivity using the same protein in either a dark or Ypet-like fusion (competent to act as a RET acceptor), and latter form showed significantly less sensitivity to light (Fig. 1i).

These results indicate engineering RET into optogenetic actuators can either increase or decrease sensitivity. In the specific case that a red fusion protein, commonly used in

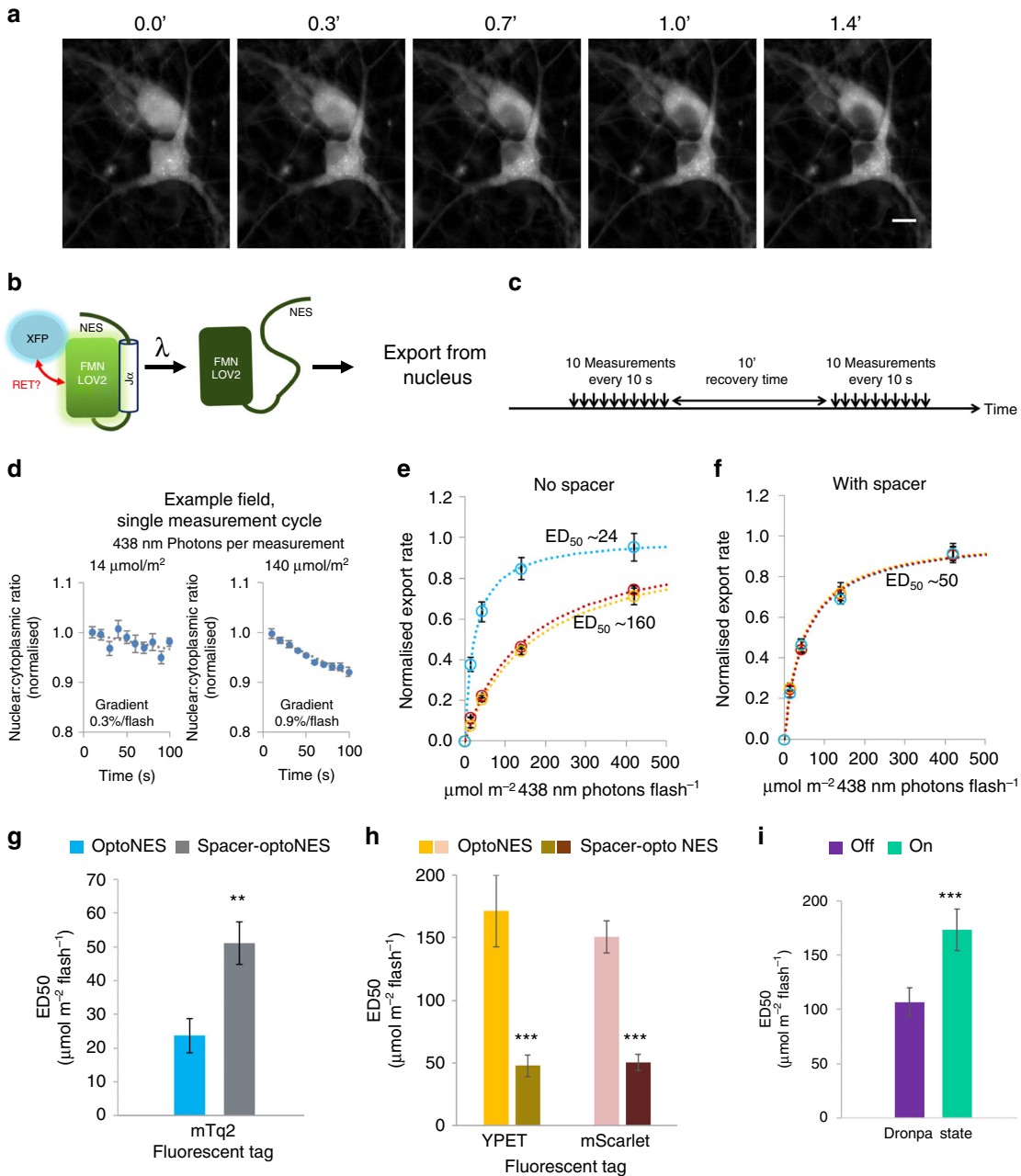

**Fig. 1 Fluorescent protein tags strongly influence the sensitivity of LOV2-based optogenetic nuclear export. a** Blue-light induces export of mTurquoise2-tagged optogenetic nuclear export construct, mTq2-optoNES (Supplementary Fig. 4) in hippocampal neurons ($n = 8$ wells at 40× magnification, scalebar 10 μm; other examples in Supplementary Fig. 5). **b** Fluorescent protein tags (XFP) fused to optogenetic constructs might influence their sensitivity to light via RET. **c** Illumination scheme used to quantify light-dependency of nuclear export rates. **d** Example data from a single 10× field treated as shown in **c** (four cycles, at least 10 min apart, mean ± SEM, for 23 and 31 cells, respectively), showing faster export at higher photon flux. **e** Nuclear export rate dependence on photon flux is shown for optoNES constructs fused to fluorescent proteins mTurquoise2, Ypet and mScarlet (in turquoise, yellow and red, respectively) expressed in different cultures. The mTq2 construct is more sensitive (significant by $F$-test, $F_{(2,66)} = 41.34$, $P < 0.0001$). Data are normalised to fitted maximal rates to facilitate sensitivity comparison (non-normalised versions in Supplementary Fig. 9). **f** Corresponding experiments with a spacer between fluorescent protein and LOV2 domain to reduce resonance energy transfer (RET, Supplementary Fig. 8). All constructs are equally sensitive to light, thus in **e**, mTurquoise had increased sensitivity whereas Ypet/mScarlet decreased sensitivity. This suggests RET can either increase or decrease optogenetic switch sensitivity. In **e-f**, all channels were imaged regardless of which XFP was used, to ensure identical illumination conditions. Data are means ± SEM, $n = 6$ wells. Best-fit $ED_{50}$s in μmol 438 nm photons.m$^{-2}$.flash$^{-1}$ indicated on the graphs for clarity and in Supplementary Table 1 with standard errors. **g** Introduction of RET-limiting spacer reduces sensitivity ($ED_{50}$) of the mTq2 optoNES >twofold; $F$-test, $F_{(1,44)} = 10.33$. **h** Corresponding sensitivities for the Ypet and mScarlet optoNES (from **e** to **f**) showing introduction of spacer to limit RET increases sensitivity >threefold; $F_{(1,44)} = 23.19$ and 49.86, for Ypet and mScarlet respectively; **i** Dronpa-optoNES shows significant difference in sensitivity in Dronpa-off and Dronpa-on states; $F_{(1, 45)} = 49.51$. **g-i** show best-fit values ± S.E. (**$P = 0.0025$, ***$P < 0.0001$) from curve-fitting of datasets in **e**, **f** and Supplementary Fig. 9F. Source data are provided as a Source Data file.

optogenetic actuators, is replaced with a cyan one, a sevenfold sensitivity increase may be achievable.

**Direct visualisation of LOV2 switching in intact cells**. RET from donor to LOV2 acceptor should quench donor emission (Fig. 2ai, left). Absorption of a photon by the ground-state (LOV-

445) leads with high probability to formation of the triplet state LOV-660[T] within 2 ns. This subsequently forms the cysteine adduct LOV-390[12], which exhibits >50% lower absorption compared with LOV-445[13]. RET to LOV2 should fall, resulting in measurable dequench of mTurquoise2 (Fig. 2ai, right). LOV-390 formation causes an immediate (~10 μs[12]) partial unfolding of *As*LOV2 flanking regions, via a glutamine lever mechanism acting

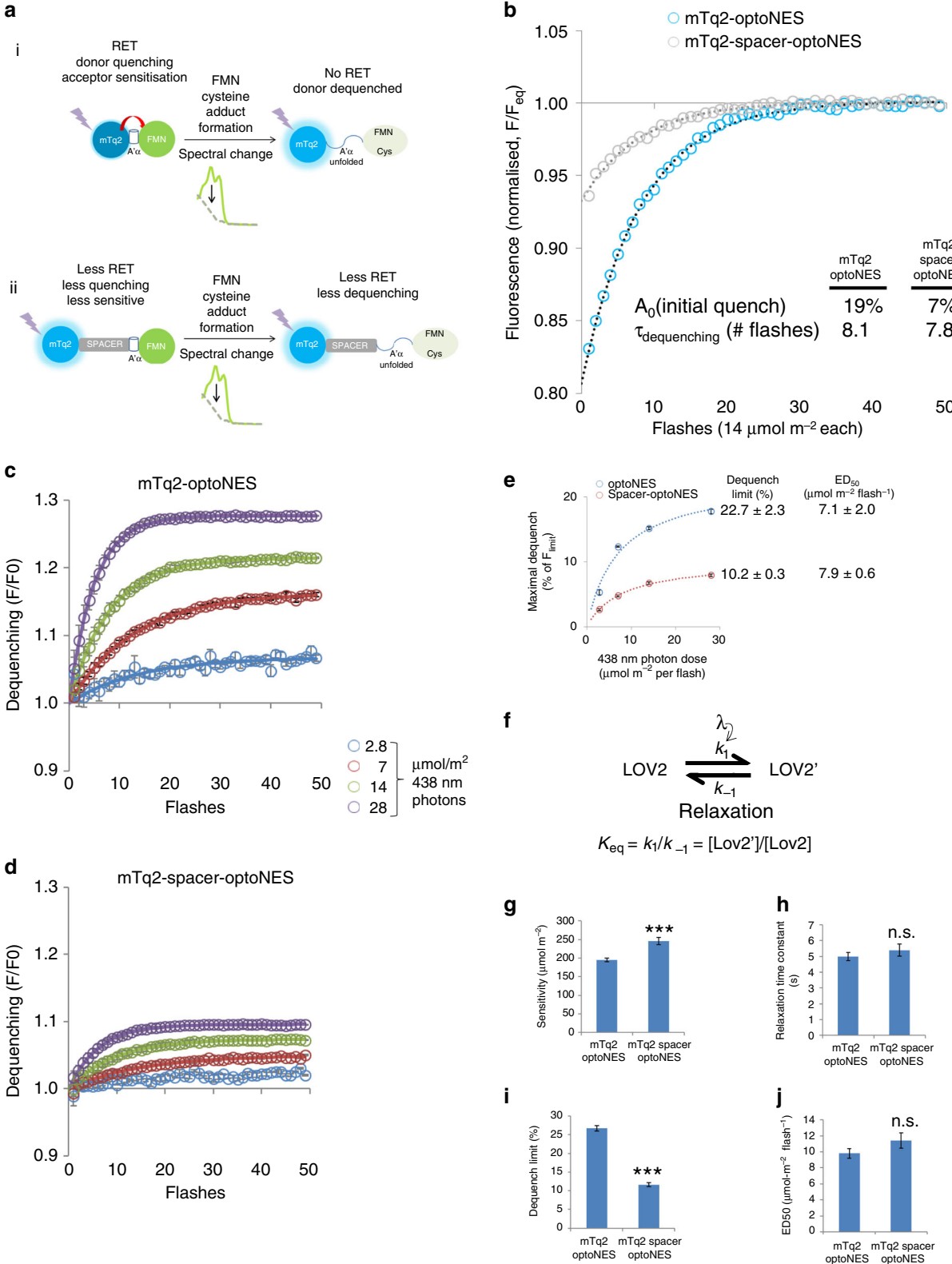

**Fig. 2 Dequenching of mTurquoise2 as a measurement of LOV2 activation. a** (i) LOV2 domain-fused mTq2 can act as a RET donor and transfer energy to FMN in the adjacent domain, and is partially quenched. Once LOV2 is activated directly or via RET, formation of FMN-adduct changes the absorption spectrum, reducing its ability to act as a RET acceptor. The LOV2 domain A'α helix extends, placing the FMN and LOV2 domain further from the fluorescent protein, also reducing RET. Thus mTq2 becomes dequenched; (ii) Placing a spacer between mTq2 and the LOV2 domain reduces RET from the outset, limiting initial quenching and the corresponding dequenching on LOV2 activation. **b** Changes in fluorescence emission of mTq2 fusion proteins, mTq2-optoNES, and mTq2-spacer-optoNES (Supplementary Fig. 8) expressed in HEK293 cells, are observed during 50 flashes of 14 μmol m$^{-2}$ 438 nm light at ~4 Hz. This replicate illustrates LOV2-dependent dequenching, with exponential fits yielding decay constants of 0.12 and 0.13 per flash i.e. τ ~8 s and differing levels of maximal dequench (or initial quench, $A_0$); **c**, **d** Means ± SEMs ($n = 3$ wells) of samples as shown in **b** were exposed to photon doses per 438 nm flash as indicated. Dequenching curves were corrected in all cases to the signal from a parallel free mTurquoise2 sample exposed to identical illumination conditions (Supplementary Fig. 10), normalised to the first data point and then plotted against flash number. Data were fitted to exponentials to derive fluorescence prior to illumination, maximum fluorescence and rate constants. **e** Data from **c**, **d**, fitted to exponentials, show the relation between maximal dequench and photon dose. Fitting to a single site model provides estimates of dequench limit i.e. the extent to which the Tq2 is originally quenched, and $ED_{50}$ the photon dose/flash to achieve half this dequenching at equilibrium when delivered at ~4 Hz. **f** All data points were fitted to a first order reversible kinetic model, providing more reliable estimates of **g** sensitivity ($1/k_1$), **h** relaxation time constant during illumination, **i** Dequench limit and **j** $ED_{50}$. ***n.s. indicate significant difference ($P < 0.0001$) and no significant difference by unpaired one-tailed multiple t-test with Holm–Sidak correction. Source data are provided as a Source Data file.

on both C-terminal Jα and N-terminal A' α helix[14]. Slower (250 μs[12]) and complete undocking of Jα follows[12,15], resulting in the order-disorder transition used to generate actuation in optogenetic tools[1]. This N-terminal A' helix unfolding reduces fluorescent tag-FMN proximity, contributing a second RET-limiting mechanism as LOV2 switches (Fig. 2ai).

As predicted, mTq2-optoNES emission increased during exposure to 438 nm light (Fig. 2b and Supplementary Fig. 10), revealing an initial quench (20% more than after equilibrium switching in intact cells; Fig. 2b, c). In contrast, the reduced-RET version (Fig. 2Aii and Supplementary Fig. 7) exhibited less dequenching (Fig. 2b–e). Dequenching is therefore sensitive to the relative placement of mTurquoise and LOV2 domains, consistent with RET. Nevertheless in both constructs, LOV2 domain-activating blue-light dequenches fused mTurquoise. Thus, dequenching curves indicate the switching of the LOV2 domain in intact cells. LOV2 switching has previously only been measured by photobleaching purified proteins in a cuvette. We can now infer the apparent sensitivity of activation from dequenching rate. Half-maximal de-quenching took ~1.3 s (5 readings, total ~140 μmol m$^{-2}$ photons). Such a fast de-quench half-time may estimate in situ activation rate if cell-free LOV2Jα relaxation rates (~20–80 s[6,16,17]) are applicable and counteracting relaxation can be neglected.

We estimated dequenching limits and activation rates more directly with rounds of activation at different photon intensities, fitting data to exponentials to extrapolate pre-illumination and final intensity levels (Fig. 2c, d). Dequenching-dose curves (Fig. 2e) provide estimates of dequenching limit (i.e. in the absence of relaxation, as photon dose approaches infinity) and $ED_{50}$ (photon dose that, at equilibrium, achieves half-maximal dequenching i.e. photoswitching). The entire data was also fitted to a first order reversible kinetic model (Fig. 2f), providing rate constants (Fig. 2g, h, for activation, $1/k_1$ ~200 μmol m$^{-2}$; for relaxation $1/k_{-1}$ ~5 s) and additional estimates of dequench limit and $ED_{50}$ (Fig. 2i, j). The activation sensitivity is comparable to observations in cells[6]. However, derived optoNES relaxation rates in situ were considerably faster than anticipated from cell-free spectroscopic measurements of LOV2Jα proteins[3,6]. This not only limits accuracy of the activation rate estimate shown in Fig. 2b but also has significant relevance for experimental use of LOV2-based optogenetic actuators.

**Visualisation and direct quantification of LOV2 relaxation.** The cysteine-adduct LOV-390 also absorbs activation wavelengths (425–475 nm). An unexpectedly short relaxation time during repeated light flashes (Fig. 2h) could potentially result from photochemical back-reactions[16]. Moreover cell-free LOV2

relaxation measurements showed a strong influence of temperature[6], and Jα modifications used to generate some optogenetic actuators[8]. We therefore sought a simple direct method to determine relaxation time in situ in darkness, as accurate knowledge of relaxation rates within experimental settings is required to design illumination protocols.

The experimental paradigm used in Fig. 2b, in which dequenching reveals the protein fraction in activated state, can be repeatedly followed by defined periods of darkness. This shows that the duration of darkness determines loss of activated fraction (Fig. 3a–c), thereby revealing the relaxation rate of the LOV2-based actuator in intact living cells (Table 1a). The experiments confirmed that optoNES construct relaxation rates are indeed shorter than values obtained by cell-free spectroscopy of LOV2Jα (tau 6–7 s vs. ~80 s[3]). The rates in darkness are slower than those derived from dequenching curves under repeated illumination (Fig. 2h), consistent with a photochemical back-reaction when LOV-390 absorbs blue light[16]. Regardless, a consistent conclusion from Figs. 2–3 is that actuators in intact cells relax much faster than expected from cell-free spectroscopy of LOV2, even when temperature is taken into account[6,16].

The activation-relaxation paradigm in Fig. 3 provides a true relaxation rate (reverse rate constant) but only an apparent activation sensitivity (forward rate constant), which underestimates the true sensitivity (Table 1a) due to the concurrent relaxation. However, based on first order reversible kinetics (Fig. 2e) the forward rate constant (sensitivity), dequench limit and photon dose required to achieve 50% maximal switch equilibrium ($ED_{50}$) can easily be calculated from fitted parameters (see methods). Therefore we primarily use the activation-relaxation approach and derived the missing parameters by curve-fitting.

**Optimisation of RET and sensitisation.** Just as RET is reduced by increasing mTurquoise-LOV2 distance, removing sequence between mTurquoise and LOV2 could increase RET. RET from mTurquoise2 doubles actuation sensitivity (Fig. 1g) and promotes LOV2 switching (Fig. 2g), allowing real-time activation/relaxation detection in living cells. We asked if RET could be further increased. A more photosensitive actuator could be switched more frequently and used for longer experiments with less phototoxicity. More sensitive detection permits more precise and frequent monitoring of actuator state.

Fluorescent protein structures reveal C-terminal residues outside the core beta-barrel structure. Deleting these residues can generate up to 95% RET in some applications[18]. We generated an mTurquoise2 deletion series lacking 7–11 C-terminal residues (Fig. 4a) fused to LOV2 residue 408. This

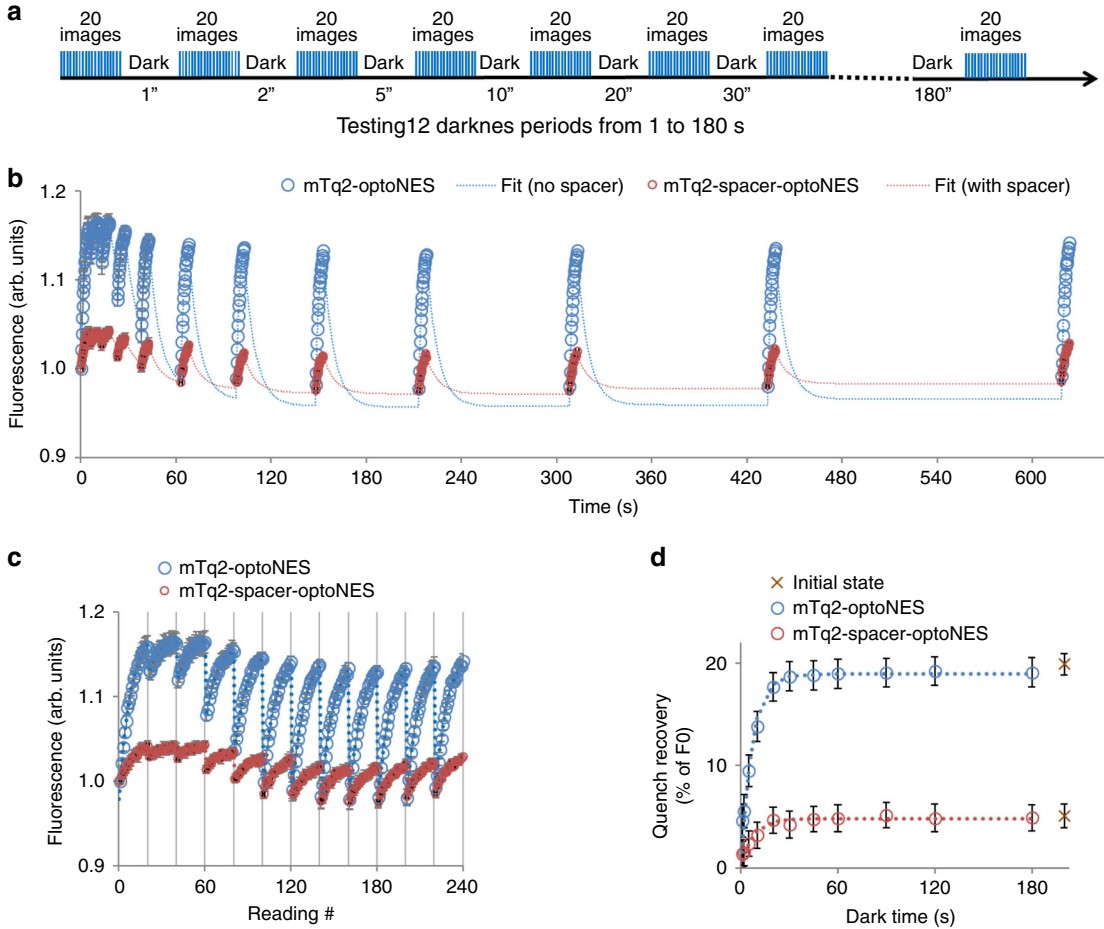

**Fig. 3 Direct quantification of relaxation time as recovery from dequenching of mTurquoise by LOV2. a** Illumination strategy to map LOV2 switching (changes in quenching) during illumination and relaxation-induced dequenching in darkness. Samples were exposed to 20 flashes (14 µmol/m² 438 nm) at ~4 Hz followed by 1–180 s darkness. **b** HEK293 cells expressing mTq2-optoNES (blue symbols) or mTq2-spacer-optoNES (red symbols) were imaged during the flash sequence as in **a** to induce dequenching followed by 12 sequentially increasing dark periods, allowing relaxation. The whole 240 image sequence was repeated three times per sample to reduce noise. Responses from each segmented cell (region of interest) in the field and from each repeat were averaged. All wellwise-averaged data from transfected cells are normalised to an average of replicates for a corresponding mTurquoise-expressing sample (without LOV2) exposed to the same illumination conditions (Supplementary Fig. 10). Means are plotted against reading number. Each dequenching phase was fitted to an exponential as in Fig. 3, with independent size and common activation rate constant for each protein, providing offset and amplitude estimates for each cycle. These curves, together with the dark-phase changes are shown as a solid line between the data points, showing goodness of fit. Apparent activation rate constants at this photon flux were 0.12 and 0.11 per flash, equivalent to a photon sensitivity of 115 and 131 µmol/ m². **c** The same data as in **b** is plotted against reading number to more easily visualise the responses. Each grid-line represents an increasingly long dark period as shown in **a**, **b**. **d** The quench recovery estimated from the exponential curve fits in **b**/**c**, is plotted against the duration of dark time causing the quench. This reveals the time-dependence of recovery from dequenching. The data is fitted to an exponential to derive relaxation time constants (~7 s for both constructs) and dequenching maxima. As evident from **b** and **c**, the dequenching is at maximum ~19 and 5% of initial fluorescence for mTq2-optoNES and the spacer construct respectively. Source data are provided as a Source Data file. Data are presented as means ± SEM; $n = 6$ wells for mTq2-optoNES and $n = 5$ wells for mTq2-spacer-optoNES.

truncates the A' helix that participates in Jα regulation but not the photocycle[17]. Confirming Jα perturbation, the NES21 motif is at least partially exposed as dC11-408LOV2-NES21 is mostly cytoplasmic (Fig. 4b), although the 408LOV2 can still optically regulate heterologous targets[19]. The deletions increase maximal dequench compared with the original mTq2-optoNES (Fig. 4c, d, Table 1a, and Supplementary Figs. 11–12). Directly fitting the entire dataset to a formula cast in terms of quench limit, sensitivity and relaxation rate, indicates the true initial quench of mTurquoise is as much as 24% (relative to LOV-390, Table 1a).

Sensitivity and dequench limit show significant correlation (Fig. 4e, Supplementary Fig. 12, and Table 2). The dC7 construct has the highest RET and sensitivity (lowest value in µmol m⁻²), suggesting further deletion either increased dipole distance or

introduced unfavourable dipole angles and RET failed to increase. Nevertheless, increased RET allows optogenetic switching with fewer photons compared with the parental mTq2-optoNES construct.

**Determinants of relaxation rate.** The relaxation mismatch between actuators in situ and reports for cell-free purified recombinant LOV2 is evident, but the cause is not. One factor is temperature[6,16], another is Jα editing[8]. The constructs in Fig. 4 have the same Jα sequence and fluorescent tag, yet in situ they exhibit threefold different relaxation rates (Table 1a). This suggests the tag-LOV2 linker can also have considerable impact. Although this potentially complicates interpretation of Fig. 1f

**Table 1 Properties of constructs examined in this study determined by resonance energy transfer.**

| Actuator | Maximal dequench or quench[1] (% of $F_{limit}$) | Apparent activation rate[2] ($\mu mol\ m^{-2}$) | Relaxation time constant in dark[3] (s) | $ED_{50}^{4}$ ($\mu mol\ m^{-2}$) | Dequench or quench limit[5] (%) | Sensitivity[6] ($\mu mol\ m^{-2}$) |
|---|---|---|---|---|---|---|
| **A** | | | | | | |
| mTurquoise2 fusions | | | | | | |
| optoNES | 15.9 ± 0.3 | 115 ± 5 | 6.8 ± 0.6 | 5.78 ± 0.22 | 18.9 ± 0.2 | 162 ± 4 |
| spacer-optoNES | 4.6 ± 0.1 | 131 ± 18 | 7.2 ± 1.1 | 6.41 ± 0.77 | 6.5 ± 0.2 | 192 ± 15 |
| dC7-408LOV2-NES21 | 25.7 ± 0.8 | 121 ± 7 | 11.5 ± 1.5 | 2.24 ± 0.29 | 23.6 ± 0.4 | 124 ± 8 |
| dC8-408LOV2-NES21 | 20.1 ± 0.6 | 134 ± 2 | 12.4 ± 1.6 | 2.29 ± 0.07 | 19.3 ± 0.1 | 137 ± 2 |
| dC9-408LOV2-NES21 | 16.9 ± 0.6 | 148 ± 4 | 13.3 ± 2.1* | 2.20 ± 0.13 | 16.6 ± 0.1 | 146 ± 4 |
| dC10-408LOV2-NES21 | 23.3 ± 0.8 | 148 ± 9 | 17.2 ± 2.3* | 1.77 ± 0.2 | 21.2 ± 0.3 | 146 ± 8 |
| dC11-408LOV2-NES21 | 21.1 ± 0.6 | 122 ± 7 | 11.0 ± 1.3 | 2.57 ± 0.33 | 20.4 ± 0.4 | 130 ± 8 |
| optoJNKi wt | 16.1 ± 0.6 | 163 ± 10.9 | 13.7 ± 1.8 | 3.7 ± 0.5 | 17 ± 0.5 | 206 ± 18 |
| optoJNKi$^{C450A}$ | n/a | n/a | n/a | n/a | n/a | n/a |
| optop38i3 | 16.4 ± 0.6 | 162 ± 5.9 | 13.8 ± 1.8 | 3.7 ± 0.3 | 17.2 ± 0.3 | 204 ± 10 |
| optop38i5 | 16.9 ± 0.6 | 163 ± 5.1 | 14.4 ± 1.8 | 3.5 ± 0.2 | 17.6 ± 0.3 | 203 ± 9 |
| Ypet fusions | | | | | | |
| optoNES | 54.5 ± 0.8 | 493 ± 9 | 8.0 ± 0.5 | 18.6 ± 1.1 | 53.5 ± 0.7 | 559 ± 12 |
| spacer-optoNES | 29.8 ± 1.1 | 430 ± 65 | 7.6 ± 1.4 | 15.9 ± 7.4 | 33.2 ± 3.7 | 479 ± 83 |
| optoJNKi | 65.6 ± 0.9 | 785 ± 4 | 23 ± 1.1 | 10.1 ± 0.1 | 60.4 ± 0.2 | 843 ± 5 |
| mScarlet fusions | | | | | | |
| optoNES | 39.4 ± 0.8 | 1140 ± 159 | 7.2 ± 0.7 | 58.3 ± 22.7 | 53.2 ± 2.1 | 1619 ± 348 |
| spacer-optoNES | 51.9 ± 0.8 | 955 ± 5 | 7.3 ± 0.5 | 44.7 ± 0.9 | 64.2 ± 0.2 | 1263 ± 12 |
| optoJNKi | 61.7 ± 1.6 | 1315 ± 33 | 15.5 ± 1.6 | 28.9 ± 1.6 | 68.8 ± 0.5 | 1594 ± 49 |

| Actuator (Ypet fusion with LOV2 mutation) | Max quench at 140 $\mu mol\ m^{-2}$ (% of $F_0$) | Apparent activation rate ($\mu mol\ m^{-2}$) at 140 $\mu mol\ m^{-2}$ | Relaxation time constant in situ (s) | Relaxation time constant reported cell-free (s) |
|---|---|---|---|---|
| **B** | | | | |
| optoNES wt | 57.4 ± 1.0 | 492 ± 5 | 7.6 ± 0.62 | 80 (Zoltowski et al. [23]) |
| optoNES V416I | 54.1 ± 1.6 | 584 ± 7 | 84.1 ± 4.7 | 821 (Zoltowski et al. [23]) |
| optoNES I427V | 28.2 ± 0.6 | 272 ± 9 | 0.81 ± 0.14 | 6 (Zayner et al. [3]) |
| optoNES F434L | 23.8 ± 1.0 | 237 ± 8 | 0.58 ± 0.25 | 12 (Zayner et al. [3]) |

Values shown are estimated best-fit ± standard error from curve fitting as described in the methods.

1. Maximal dequench or quench—this is the equilibrium impact on emission from the fluorescent protein on repeated illumination with 438 nm light (cf Fig. 2e). It is caused by resonance energy transfer to or from the nearby LOV2, but the equilibrium point depends also on continuous relaxation. For this reason the value depends on the rate (~4 Hz) and intensity of 438 nm light flashes, which is indicated. mTurquoise2 fusions were dequenched at 14 $\mu mol\ m^{-2}$ whereas Ypet and mScarlet fusions were less sensitive and were quenched at 140 $\mu mol\ m^{-2}$.
2. Apparent activation rate is the measure of apparent sensitivity of switching, again influenced by concomitant relaxation and dependent on photon flux, which is indicated for each group. mTurquoise2 fusions were activated at 14 $\mu mol\ m^{-2}$ whereas Ypet and mScarlet fusions were less sensitive and were activated at 140 $\mu mol\ m^{-2}$.
3. Relaxation time in darkness is calculated from periods of darkness and is independent of activating photon flux. In Table 1B, the measured values of relaxation mutants in the optoNES context are compared to literature values of recombinant cell-free LOV2 mutant without actuator or fluorescent protein fusion.
4. $ED_{50}$ is the estimated photon dose at the rate used (~4 Hz) predicted to achieve half-maximal switching at equilibrium, taking concomitant relaxation into account.
5. Dequench or quench limit—this is the estimated relaxation independent of maximal possible quench (for acceptors Ypet and mScarlet) or dequench (for mTurquoise2).
6. Sensitivity is the estimated sensitivity value in the absence of relaxation derived from curve fit formulae that take the concomitant relaxation rate into account.
An asterisk denotes statistically significant difference in relaxation between optoNES and dC9 ($P = 0.0364$) and dC10 ($P = 0.0004$), and spacer-optoNES and dC10 ($P = 0.001$) constructs by one-way ANOVA and post-hoc Tukey test. No other significant difference was found among the mTurquoise-optoNES construct in situ relaxation rates.

where RET is limited with a spacer, Fig. 2 showed the spacer used did not affect relaxation rates. Whether the cellular environment also influences relaxation is unclear. We compared in situ relaxation rates with those determined in cell-free HEK293 lysates at different temperatures (Fig. 4f). This confirms the impact of temperature but, for the actuators tested, shows no evidence for contributions of the cellular environment per se (Fig. 4f).

**In situ photocycle monitoring generalises to other actuators.** Could activation-relaxation cycling report in-cell sensitivity and relaxation of other optogenetic actuators? We selected optoJNKi, an optimised JNK inhibitor design that revealed resonance in JNK signalling circuits[6] and JNK regulation of dendritic spines[20], and similar p38 inhibitors, optop38i3 and optop38i5[6]. All activatable constructs generated dequench-recovery cycles (Fig. 5a, b). The LOV2 mutant optoJNKi-C450A (aka dark state mutant, or optoJNKi.dsm[6]) binds FMN but lacks the cysteine that forms the FMN-adduct and cannot photocycle[21]. No FMN-adduct can be formed and thus RET should not change. We observe no cyclic dequenching-requenching (Fig. 5a).

Relaxation times are again shorter than expected from cell-free spectroscopy (Fig. 5c and Table 1a), even comparing to His-tagged optoJNKi at 37 °C[6], consistent with an influence of N-terminal linker. These constructs have intact Jα (to residue 546) unlike optoNES and LEXY (to residue 541 only), and have longer relaxation times, consistent with Jα modification hastening

relaxation[8]. The best-fit dequench and sensitivity parameters for these actuators map above the sensitivity-FRET (dequench limit) relation of optoNES constructs with modified Jα (Fig. 4e and Supplementary Fig. 12), again consistent with a stable Jα.

**Photocycle monitoring generalises to other fluorophores.** In situ determination of activation and relaxation rates requires a RET-competent tag like mTurquoise (Figs. 2–5). Figure 1 shows RET also occurs with yellow and red proteins, and presumably any protein having spectral overlap with LOV2. Substituting mTurquoise with Ypet showed a larger RET effect (Fig. 6; quench limit up to 66%, Table 1a), consistent with greater spectral overlap (Supplementary Fig 1). As LOV* acts as a donor here, Ypet emission upon 438 nm excitation is initially enhanced (approximately twofold, Fig. 6a) but each flash converting LOV-445 to LOV-390 reduces Ypet emission, consistent with RET loss, thus traces are inverted compared with Figs. 2–5. A dark period recovers the sensitised Ypet signal. Activation and relaxation parameters can be obtained from Ypet-LOV2 fusions (Fig. 6e–g and Table 1a). RET from LOV2 to Ypet causes loss of energy from S1-state LOV* and activation sensitivities are >fivefold lower than mTurquoise2 versions. The RET-limiting spacer (Fig. 1 and Supplementary Fig 8) reduces dequench limit. This is the opposite of the mTurquoise2 constructs, consistent with Ypet acting as acceptor. Red protein mScarlet has similar effects to Ypet (Supplementary Fig 13 and Table 1a). This is predictable

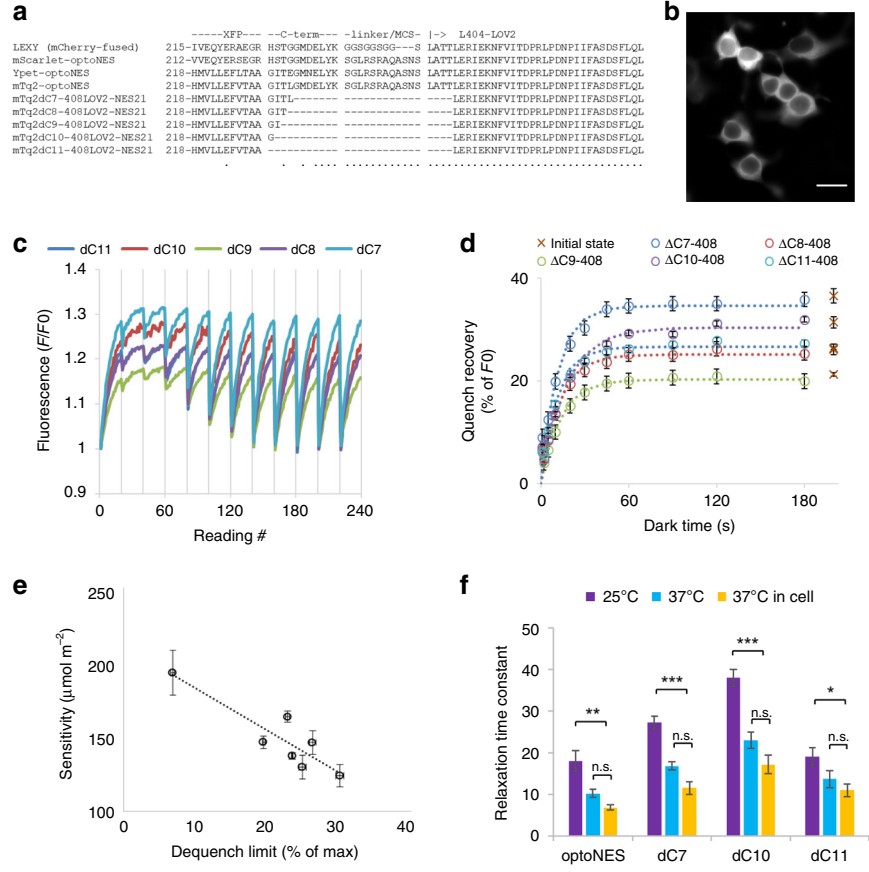

**Fig. 4 Increased RET to LOV2 and sensitisation can be achieved by linker optimisation. a** Alignments of XFP-optoNES variants compared with the original mCherry-fused LEXY[9], showing the end of the fluorescent protein βbarrel and intervening sequences. *As*LOV2 starts at leucine 404 or trimmed to 408 to increase RET; **b** The trimmed constructs are typically constitutively cytoplasmic because disruption of A'α helix by deleting LOV2 residues 404–407 renders the constructs functionally active as expected (the dC11 construct is shown; scalebar 20 μm, $n = 3$ wells in ×10 magnification, 1 example imaged at ×40); **c** The C7-C11 deletion series of mTq2 fused to 408LOV2-NES21 were subjected to kinetic RET analysis, revealing characteristic dequench-relaxation cycles (mean, $n = 3$; SEM shown in Supplementary Fig. 11); **d** Analysis of the dequench-relaxation as in Fig. 3 reveals that deletion of linker can almost double maximal dequench (at 14 μmol m$^{-2}$) though relaxation rate is also affected. The dC7 construct has least deletion but greatest dequench, suggesting dipole angles or other parameters reduce RET in other constructs though mTq2-LOV2 distance in primary sequence is reduced. Best-fit parameters—maximum dequench, apparent sensitivity, relaxation time and ED50, dequench limit and sensitivity are compared in Table 1A. **e** Activation sensitivity of the mTurquoise-optoNES constructs from Table 1A correlates with dequench limit (equivalent to effective FRET efficiency; statistical analysis in Supplementary Table 2, best-fit ± S.E., $n = 3$ wells). **f** Best-fit relaxation rates obtained from the in situ data (green bars; Table 1A) are shown together with the corresponding values derived from measurements performed on lysates of cells expressing the same constructs (dC7, dC10, dC11 and the parental optoNES) at 25 and 37 °C (blue and red bars). This shows that temperature has a strong influence on relaxation rates of these constructs but, when the sequences used are identical, there is no significant difference between relaxation in intact cells and the cell lysates in any case. Mean ± standard error is shown ($n = 8$ cell-free, $n = 6$ and 3 for in-cell wild-type and variants respectively). *, **, *** denote $P < 0.05$, 0.01, and 0.0001, respectively (25 °C compared to either of the 37 °C values) by Tukey post-hoc test after two-way ANOVA. n.s. not significant. Source data are provided as a Source Data file.

given the spectral overlaps (Supplementary Fig. 1) and the longer wavelength proteins acting to limit LOV2 responsiveness. Dark state relaxation times, a thermal process of adduct decay, are generally insensitive to the fused fluorescent tag (Table 1a), in contrast to the strong effects of the linker sequence. However exceptions exist, as Ypet-fused optoJNKi has an unusually long relaxation time.

**Parameter-based minimisation of light dose for actuation.** The parameters applied allow prediction of activated state dynamics of the actuator during illumination trains, even if it does not translocate or provide visual cues of activation. The optoJNKi fraction in adduct state (LOV-390) reduces rapidly after each activating light flash (relaxation time-constant of 14–23 s, Table 1a), so frequent illumination is required to maintain activated optoJNKi. The imaging-based translocation reporter JNKktr[22] allows us to determine JNK inhibition during a given

illumination pattern. We aimed to efficiently stimulate optoJNKi by repeated illumination while simultaneously quantifying inhibition of the JNK pathway in multiple samples. Sequential periodic epi-fluorescence imaging in multiple samples was paired with parallel optoJNKi activation by a multiwell trans-illuminator device (Fig. 7a). We predicted adduct state levels over time (Fig. 7b), based on activation and relaxation parameters determined for mTq2, Ypet and mScarlet-optoJNKi (Table 1a). Extremely low photon fluxes (60 μW cm$^{-2}$ = 2.3 μmol m$^{-2}$ s$^{-1}$, for 1 s every 7.5 s) were selected to avoid detectable long-term effects on gene expression or toxicity even under media-enhanced phototoxic conditions[2]. The peak adduct state fraction was predicted to be <3% before each decay in the case of mTq2-optoJNKi (Fig. 7b). Once LOV2-445 is regenerated, there will be a delay before the construct dissociates and the target becomes available for interaction with substrates. The adduct state fraction is therefore an underestimate of the optogenetic inhibitors capable

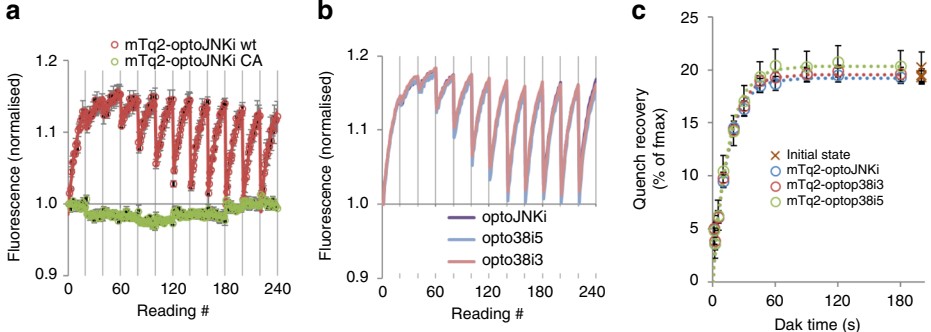

**Fig. 5 Apparent activation rates, dark state relaxation rates and calculated sensitivities of selected Lov2-based actuators fused to mTurquoise2. a** The optogenetic JNK inhibitor optoJNKi fused to mTq2 is evaluated using light-dark cycles as in Figs. 3-4 and compared with a C450A mutant version (green) that cannot form a cysteine adduct and therefore cannot cycle, as shown. Means ± SEM ($n = 3$ wells) are plotted together with a series of exponential fits with a common activation constant (red line); **b** mTq2 fusions of optoJNKi, optop38i3 and optop38i5 are evaluated and the response curves show almost complete overlap. Raw averaged data ($n = 3$ wells) is shown for each construct; **c** Best-fit amplitudes (±S.E.) of exponentials fitted to the data in **b** are plotted against preceding dark time; crosses plotted nominally at 200 s represent the amplitudes of the first exponential which was not preceded by any light exposure. These data are fitted to exponentials to derive the relaxation time in dark and maximal dequench (as % of initial signal) for each construct under the low illumination conditions used (14 $\mu$mol m$^{-2}$ flash$^{-1}$ at ~4 Hz). Max dequench (as % of total), apparent activation rates, and relaxation time constants in dark for the data in **b**, **c** are shown in Table 1A. From these values, ED$_{50}$, dequench limit (equivalent in each case to FRET efficiency, assuming adduct has no effect) and sensitivity are derived. The C450A optoJNKi construct, which cannot form an adduct and therefore for which no activation or relaxation values could be derived, is also listed in the table. Source data are provided as a Source Data file.

of binding targets. Nevertheless a clear difference between mTurquoise2, Ypet and mScarlet versions is predicted as we likely avoid saturation of targets. These illumination conditions caused significant light-dependent inhibition by mTq2-optoJNKi, minimal light-dependent effect of Ypet-optoJNKi and no discernible impact of light on mScarlet-optoJNKi infected neurons (Fig. 7c–f).

This demonstrates the generality of the optoNES findings (Fig. 1), that sensitivity of optogenetic actuation under linear illumination conditions can be modulated by fluorescent fusion tags. It also demonstrates that in situ RET-derived activation and relaxation parameters aid the selection of minimal illumination conditions sufficient for switching (and hence actuation). This means that, when using optogenetic actuators, the blue-light toxicity artefacts can be limited even in long-term studies, and the likelihood of achieving intended outcomes is increased even where on-line validation assays are not available.

**RET guides optimisation of in situ relaxation time.** Relaxation of all optogenetic constructs tested in situ is consistently faster (up to tenfold) than for recombinant LOV2 measured by absorbance spectroscopy. This mismatch complicates the application of cellular optogenetics. For example, the high-frequency (0.13 Hz) illumination required (Fig. 7), if delivered through-the-objective, greatly limits throughput. Parallel trans-illumination devices (Fig. 7a), in providing an illumination path orthogonal to the epifluorescence microscope optical path, can resolve the sample throughput bottleneck. But rapid relaxation time also means higher blue-light doses are required to maintain switching (and hence actuation).

For these reasons we turned to relaxation-time mutants of LOV2. The impacts of mutating V416, I427 and F434 and others on decay rate of the FMN adduct and therefore relaxation times have been reported by cell-free spectroscopy[3,23], and in some cases impact on downstream actuation has been quantified in intact cells[24,25]. However the effect of these mutations on actual state-switching of LOV2, independent of actuation-specific complexities, have not previously been measured in situ. The LOV2 FRET-based in situ switching assay shows that LOV2-V416I slows Ypet-optoNES relaxation rate approximately tenfold

whereas I427V and F434L mutants both show ~20-fold faster relaxation (Fig. 8 and Table 1b). These fold-changes are consistent with spectroscopic data using *As*Lov2 without Jα modification, but optoNES in situ time constants are 10- to 15-fold faster[3,23].

**Re-programming and quantifying signalling pathway dynamics.** A slower-relaxing optoJNKi would allow simpler and more efficient experimentation than in Fig. 7. mTq2-optoJNKi-sr (slow relaxation), incorporating the V416I mutation, has a relaxation time of ~2 min (Supplementary Fig. 14), 7 times slower than the original optoJNKi (Table 1a) and is the most sensitive construct tested (lowest ED$_{50}$). We compared the applicability of optoJNKi variants to regulate neuronal JNK pathway activation by paclitaxel and vincristine, clinically used chemotherapeutic agents associated with central neurotoxicity[26–28].

In a traditional through-the-objective actuation configuration —where imaging and actuation light use a common epifluorescence path—optoJNKi requires five flashes per min to suppress JNK activity (Fig. 9a–c), precluding parallel sample imaging. Response modulation is gradual (tau ~4–16 min), demonstrating the requirement for continued inhibition to modify the response. In contrast, 0.5 flashes per min is sufficient for optoJNKi-sr to rapidly suppress the JNK pathway (Fig. 9d, e). This permits multi-sample imaging, >tenfold increased throughput, and JNK inhibition with tenfold less blue light. Under these conditions, neurons tolerate repeated long-term modulation of the pathway (Fig. 9e).

Selection of a LOV2 mutant with in situ response times matching target kinetics (JNK regulation) simplifies optogenetic modulation of the JNK pathway. Although JNKktr requires nuclear-cytoplasmic translocation to report a change, measured translocation rates (tau 1.8–2.8 min[22]) are considerably faster than response rates seen here (Fig. 9f). Light-evoked inhibition of reporter is threefold faster than recovery in darkness 20 min later (Fig. 9d, f). This suggests phosphatases act more rapidly than kinases on the c-Jun derived reporter in this condition, and reporter translocation cannot explain this discrepancy. In Fig. 9e, JNK is inhibited considerably longer and the difference is greatly diminished (Fig. 9f). This reveals significant differences in re-

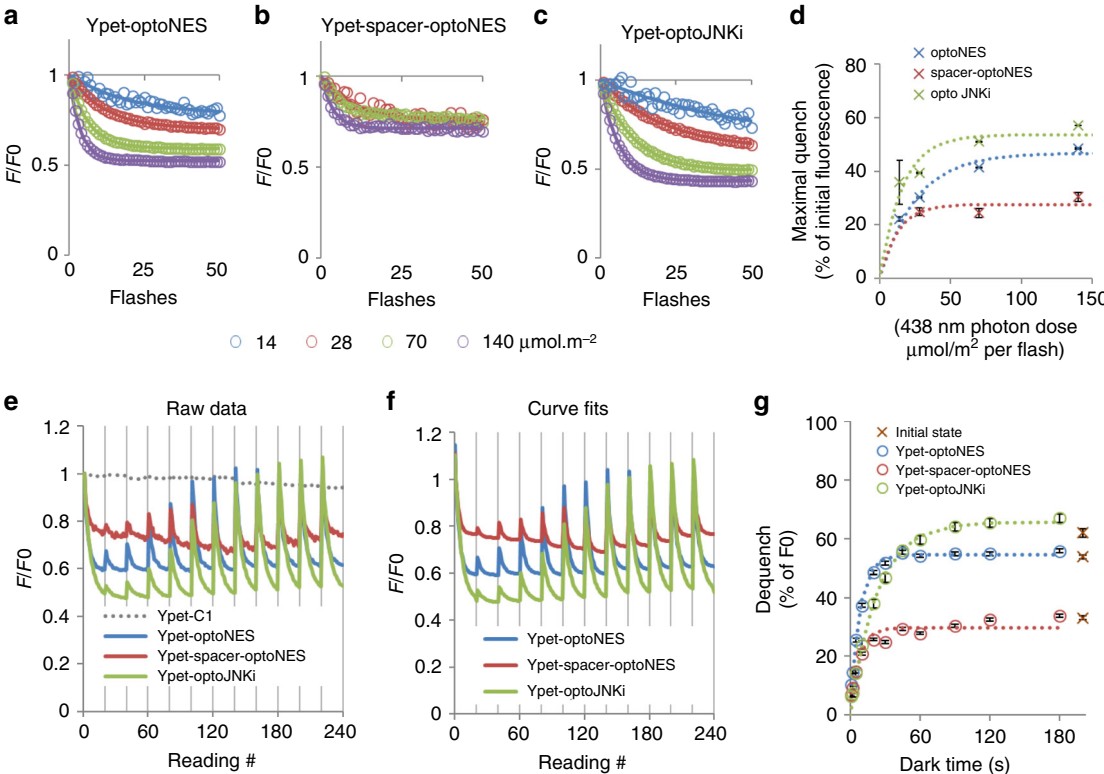

**Fig. 6 Activation rates and dark state relaxation rates of LOV2-based actuators fused to Ypet. a** Fluorescence emission of Ypet-optoNES, detected through a 542/26 nm filter, on excitation pulses with 14–140 μmol m$^{-2}$ of 438 nm excitation light (as Fig. 3) shows reducing intensity or quenching at a rate depending on the photon dose. This is consistent with a FRET from LOV2* to Ypet that is diminished as LOV2-390 is formed. Note 5x more light is used here than for mTq2 fusions. Similar results are obtained with **b** Ypet-spacer-optoNES and **c** Ypet-optoJNKi. Means ($n = 3$) are shown. **d** shows the best-fit maximal quench increases (±S.E., $n = 3$ wells) with illumination intensity increases, allowing a quench limit to be estimated by fitting to a single site model (dotted line). Quench limits, activation rate constants and relaxation time constants during illumination for the Ypet constructs, and corresponding data from mScarlet constructs, are shown in Supplementary Fig. 13, indicating FRET from LOV2 to the fluorescent tags in all cases and to a reduced extent in the spacer construct (the Ypet-spacer construct with limited energy transfer was too dim for parameter estimation). **e** The constructs (means of $n = 3$ wells shown) were subjected to cycles of activation and darkness as in Fig. 4 and fluorescence (438 nm excitation, 500 nm emission, corrected to control Ypet-C1 signal shown as dotted line) measured and **f** fitted to repeated exponential decays with a common time constants for each construct in each experiment; **g** The best-fit dequench ± S.E. from **f** is plotted against the preceding dark time and fitted to an exponential to derive a relaxation time in darkness. The best-fit parameters—max. quench at 140 μmol m$^{-2}$ photons, apparent activation during concurrent relaxation and relaxation time during darkness—estimated from activation-relaxation cycles are listed for both Ypet and mScarlet fusions, together with derived parameters, in Table 1A. The quench limits and sensitivities are in general consistent with **d** and dark relaxation times are slightly slower than lit relaxation times, suggesting a contribution of photochemical back reaction in **d**. Source data are provided as a Source Data file.

activation rates between the two conditions but no difference between inactivation rates (Fig. 9f). This could reflect negative feedback on the JNK pathway, whereby persistent inhibition of substrate phosphorylation by optoJNKi-sr in Fig. 9d results in higher activity, leading to more rapid recovery of the reporter response in Fig. 9d. This data alone does not discriminate between changes in kinase or phosphatase activities. Subsequent recovery in all cases to about the same level (as prior to inhibition) argues either for balanced changes in both or a rapid return to the initial balance, but more experiments would be needed to distinguish these possibilities. Nevertheless, this shows tools like optoJNKi can be optimised to repeatedly inhibit and release a target in an efficient contact-free manner with minimal illumination. Traditional inhibitor approaches do not allow this longitudinal approach, multiple parallel samples would be required. Moreover penetration, distribution, washout and turnover rates of inhibitors, together with the stressful effects of cell washing, introduce further complications. In contrast, optoJNKi-sr permits a highly versatile and repeatable in situ probing of pathway dynamics from single samples at multiple time-points during a response.

## Discussion

Optogenetic approaches have revolutionised behavioural neuroscience. Implementation in cell biology and related fields has also been impressive but limited to relatively few labs. It remains challenging to develop and use optical regulators of biochemical processes. One difficulty is the inability to directly monitor the switched state of optogenetic actuators. The vast majority of cellular optogenetic actuator studies generate easily visualised end points, such as protein translocation and cell motility[9,24,29–32], where cellular outcomes can be directly verified. This is a powerful approach when activation of signalling is achieved by changes spatially tethered to a membrane compartment, for example[24,33,34]. However, many targets are not constrained to easily visualised loci. This problem is compounded by invisible spontaneous relaxation of actuators to the inactive dark-state. Relaxation rate has been measured in cell-free conditions but not in intact biological systems. Optogenetic inhibitors of endogenous delocalised activities—the optogenetic equivalent of pharmacological tools—are particularly rare because indirect methods are generally required to validate their actions. Optogenetic activators can be straightforward to implement for proof-of-concept in

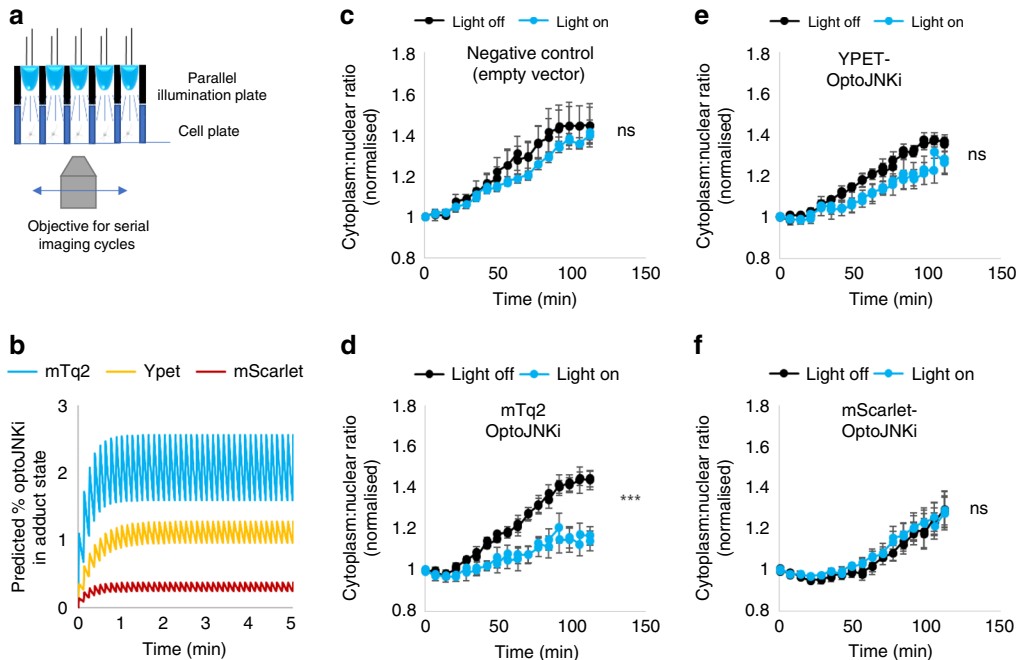

**Fig. 7 Minimal illumination conditions sufficient to inhibit the JNK pathway inhibition with mTq2-optoJNKi and demonstrate fluorescent tag-dependent sensitivity of actuation. a** To evaluate activation of the JNK pathway in real-time, multipoint epi-fluorescence microscopy of neurons expressing JNK reporter was carried out during trans-illumination of selected wells by blue LEDs gated by microscope imaging acquisition. **b** Based on parameters from Table 1, and a 1 s 0.06 mW/cm$^2$ flash of blue light every 7.5 s, the % of construct in adduct state stabilises after about 1 min as shown. The mTq2 construct reaches approximately fivefold greater proportion in adduct state compared with the Ypet construct, whereas the mScarlet construct exhibits minimal adduct state under these conditions. **c–f** Changes in cytoplasm:nuclear ratios of the JNK pathway reporter JNKktr (increased translocation from the nucleus indicates activation of JNK) are shown in response to treatment with JNK-activator anisomycin. Cultured cortical neurons were infected with AAVs encoding synapsin-promoter driven JNKktr fused to miRFP670 and optoJNKi fused to either mTq2, Ypet or mScarlet as shown. The light-on samples (turquoise datapoints) were exposed to light as shown in **a**. The mTq2-optoJNKi expressing neurons exhibited significant light-dependent inhibition, the Ypet-optoJNKi had a barely detectable and not significant effect in response to light whereas the mScarlet-optoJNKi expressing cells did not respond to light. This is consistent with the predictions shown in **b**. Means ± SEM (n = 4 wells) are shown. ***P < 0.001, ns denotes non-significance, by two-way ANOVA. Source data are provided as a Source Data file.

model cell lines. However, inhibitors of endogenous signalling processes can be more valuable tools to investigate physiological mechanisms than imposition of overexpressed synthetic activity. Moreover, biological end-points of interest may require maintenance of actuation for prolonged periods of time, unlike the milliseconds of activation in channelrhodopsin applications. For popular blue-light photosensors, using endogenous flavins rather than exogenous chromophores like phytochromobilin, this is an additional challenge because even low doses of blue light can perturb biological systems in the long term[1,2].

Building on the easily visualised responsivity of LEXY[9], the light-regulated NES, we found sensitivity could be doubled by engineering RET from a cyan fluorescent donor (Fig. 1e). Conversely red and yellow proteins act as RET acceptors, reducing sensitivity ~3.5-fold (Fig. 1 and Supplementary Table 1). Sensitivity can be dynamically tuned with a photo-switchable protein. This potentially opens the door to multiplexing, whereby orthogonal inputs (non-blue light) determine which actuator present will respond to blue light. Linker optimisation increases energy transfer from the cyan protein (Fig. 4 and Table 1a), potentially further sensitising actuation (Fig. 4e). The finding that fluorescent tags modulate activation under linear illumination complements previously described mutational tuning of LOV2 relaxation time[3,35]. Combining control of both activation and relaxation rates offers precise blue-light activated cellular optogenetic design. Long-term experiments, with considerably more optogenetic manipulations, become possible before blue-light phototoxicity becomes limiting. If fewer photons are required to

activate LOV2, improving temporal resolution with faster relaxing LOV2 mutants becomes possible.

Sensitisation of flavin-based optogenetic switches under non-linear illumination (two-photon microscopy) was recently described[36] using mtagBFP and mTFP1 for CRY2 and LOV2 respectively. Sensitisation was not seen in wide-field mode, perhaps because mtagBFP2 and mTFP1 are less suitable than mTurquoise2 for FRET. We predicted limited overall RET between mTFP1, with characteristics mid-way between mTurquoise2 and Ypet, and LOV2 (Supplementary Fig. 15A, B). In practice, mTFP1 performs poorly as an indicator of AsLOV2 state compared with mTurquoise2 or Ypet (Supplementary Fig. 15C–E). Under two-photon microscopy, two photons are absorbed within a short time, and the higher absorbance of mTFP1 than AsLOV2 becomes the dominant factor under non-linear conditions. Under more commonly used linear excitation, we found mTurquoise2 offers at least twofold increased optogenetic sensitivity.

A second consequence of RET in LOV2-based tools is that the photocycle, i.e. optogenetic actuator switching, can be generically monitored in situ because this influences fluorescent tag quenching. Note this does not report the context-specific state of the downstream actuation machinery and its interaction with targets in the cellular environment. Nevertheless these depend fundamentally on the optical switch state, which is easily measured. Three alternative but less flexible methods were described earlier to monitor generation of the activated state of LOV2 in real-time. First, the weak fluorescence of resting-state LOV2 itself was directly measured in a bacterial screen[37]. The low

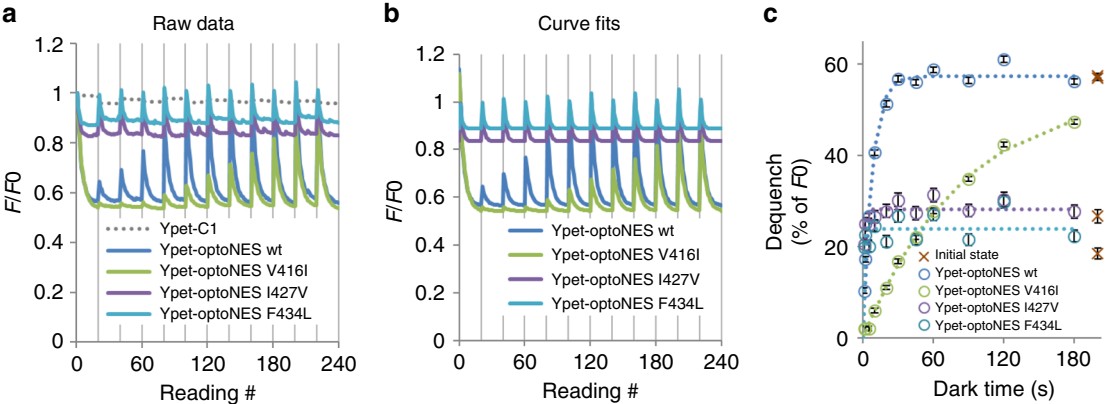

**Fig. 8 RET allows mutational relaxation time optimisation to be monitored in situ. a** Raw (vector-corrected) fluorescence quench-recovery cycles captured from intact living HEK293 cells using the protocol used in Figs. 2–6 for quantitation of LOV2 switching and relaxation of optoNES constructs with wild-type LOV2 or photocycle mutants V416I, I427V and F434L. These are tagged with Ypet because the LOV2 FRET responses are larger and clearer. The dotted line shows the changes in the Ypet vector that does not contain LOV2, which is used to correct all traces. Data are means from $n = 3$ wells. **b** Curve-fits of vector-normalised quench-recovery data from **a** are shown. Curve fitting is performed as described in the methods and provides activation time constants and dark dequench values plotted in **c**. **c** Relaxation time constants are estimated by exponential fits of the % of dequench values from **b**, and best-fit values (±S.E., $n = 3$ wells) plotted against dark time preceding the dequench for each construct. The red crosses represent the initial dequench value of the first measurement (taken as representing the dark state because the cells had not previously been exposed to light). Averaged parameters based on the experiments in parts **a–c** are listed in Table 1B. Relaxation time constants for the fast photocycle mutants I427V and F424L are 10–20 times shorter than the wild-type and V416I is about 10× longer. Relaxation is a consistent 10–15-fold faster than has been reported under cell-free conditions[3,23]. The extents of activation of the fast relaxing mutants are low, because of the considerable concurrent relaxation taking place during the activation conditions used. For the same reason apparent activation sensitivity and maximum dequench change inversely with relaxation rate. This is expected because a slower relaxation allows a more complete activation during the illumination phases while faster relaxation counteracts apparent activation and reduces the possible level of dequench at limiting photon flux. Source data are provided as a Source Data file.

fluorescence quantum yield (0.13 at room temperature) quenched by elevated temperature[38,39], low extinction coefficient and broad spectra overlapping with multiple channels (Supplementary Fig. 1) make LOV2 fluorescence inadequate for single-cell imaging in practice. LOV2 is theoretically ~50-fold dimmer than Ypet, making it sensitive to bleed-through errors in multichannel experiments. Second, the conformational change in AsLOV2Jα and optogenetic CaMKII inhibitor paAIP2 modified by fusion to GFP and dark RET-acceptor ShadowY at N and C-termini generate detectable FLIM change[7,40]. However, this approach may not be generally applicable; C-terminal fusion may inactivate inhibitors requiring free C-termini (e.g. PDZ domain ligands[41]) or when docking of the C-terminal residue of the inhibitor with a LOV2 pocket is important for photo-regulation[6]. Moreover, functionality of the modified construct used to demonstrate photoactivation was unclear as it was not used for inhibition[7]. Third, Murakoshi et al. (ref. [7]) co-expressed Cherry-paAIP2 with a GFP-fused version of the inhibitor target, CaMKII in this case, to achieve a FLIM readout. This may not be routinely applicable as (i) successful two-component RET is dependent on distance and angle parameters not easily controlled, (ii) target over-expression is usually undesirable in inhibitor experiments, and (iii) the targets of optogenetic inhibitors are usually endogenous signalling components that are not fluorescent. Moreover, our results suggest that both EGFP and ShadowY-fused constructs might, like Ypet fusions, exhibit RET and reduced sensitivity. Our more generically applicable and simpler dequenching method, requiring only a conventional microscope, could be a routine alternative to these more technically challenging approaches.

A third consequence of RET with LOV2-bound FMN is the in situ determination of relaxation time. Relaxation is typically assumed a fixed LOV2 domain property albeit altered by mutation[3]. Relaxation is faster at 37 °C in cell-free spectroscopic studies[6,16], but actuators' relaxation measured in situ appeared even faster. Jα modification has significant effects[8], whereas the

photoregulated peptide has a modest impact[6]. Here we demonstrated that the fluorescent tag usually has minimal impact but the linker between tag and LOV2 has a strong effect. We conclude that both temperature and the actuator sequence, both within the LOV2-Jα and the N and C-terminal flanking sequences, each contribute to define the relaxation time. It is important to quantify relaxation rates of optogenetic actuators used, but these are easily determined by LOV2FRET, either in situ or in cell lysates. The former may have greater accuracy and experimental relevance. However, the cases we compared showed no evidence for an impact of the intracellular environment per se on relaxation time.

In situ activation and relaxation parameters predict illumination requirements for actuation. The availability of in situ JNK reporters allowed us to demonstrate this in practice with an optoJNKi panel fused to different fluorophores (Fig. 7). However, the straightforward calculation algorithm used (Supplementary file) guides design of effective illumination sequences even when in situ reporters are unavailable and on-line validation not feasible.

Fast relaxation complicates multi-point objective-based imaging and increases blue-light dosage. The *As*Lov2-V416I mutant slows relaxation approximately tenfold[3,23] and has occasionally been implemented in optogenetic actuators[24,25]. Used in optoJNKi, the in situ relaxation rate is still sufficient for investigating the slow kinetics of the JNK pathway, while providing tenfold reduced light exposure and increased experimental throughput without loss of temporal precision. This allowed us to show that transient suppression of a pathway offers repeatable and relatively non-invasive in situ quantification of pathway dynamics, during both inhibition and re-activation not achievable by pharmacological approaches.

In summary, RET can sensitise an optogenetic switch >twofold (sevenfold when replacing a red or yellow protein). RET also reports real-time in situ activation state, revealing activation sensitivity and

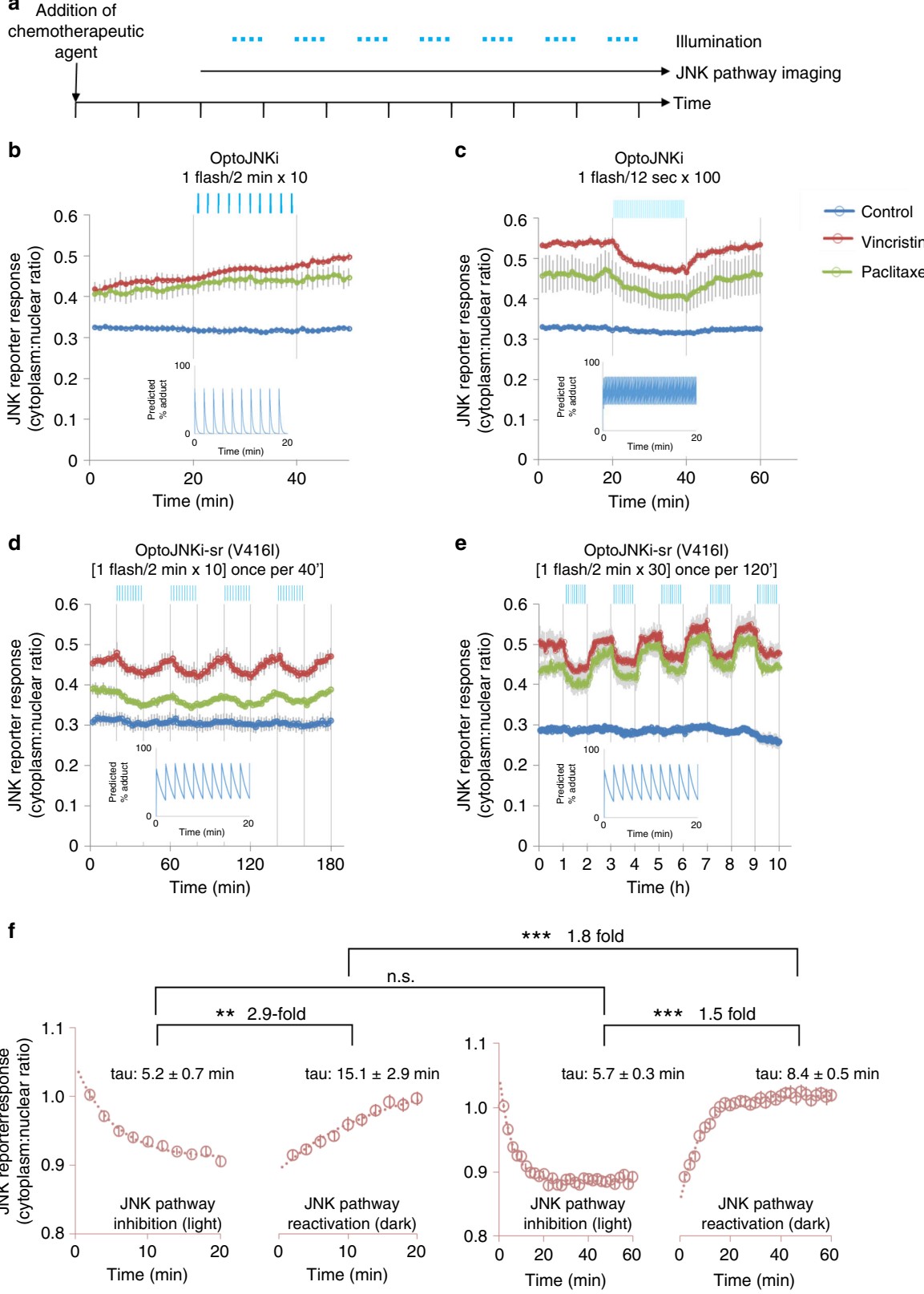

relaxation rates. These parameters guide selection of optical conditions and implementation of mutants, reducing photon dose up to tenfold while increasing experimental throughput tenfold. Finally, this improved experimental throughput allows quantification of changes in bidirectional pathway kinetics in intact living cells over time.

## Methods

**Reagents and plasmids**. Anisomycin was purchased from Sigma-Aldrich, and Vincristine and Paclitaxel from Adooq. Dulbecco's Modified Eagle Medium, Foetal Bovine Serum, Neurobasal™-A medium and B-27™ supplement were from ThermoFisher Scientific. Other reagents were obtained from Worthington, Sigma, Applichem, Apollo Scientific, NEB, ThermoFisher Scientific, Takara and VWR. PCR templates for mScarlet, miRFP670 and Dronpa were generous gifts of

**Fig. 9 Optimisation of relaxation in optoJNKi-sr increases sensitivity and experimental throughput tenfold, facilitating investigation of bidirectional pathway dynamics in intact living cells. a** Neurons expressing mTq2-optoJNKi and miRFP670-JNKktr were exposed to chemotherapeutics (Vincristine and Paclitaxel, 100 nM each). The JNK pathway was measured during alternating periods of blue light flashes and darkness; **b** After 6 h, JNK reporter activity increases steadily. Using the original optoJNKi, blue light (420 μmol m$^{-2}$ 438 nm every 2 min, 8.3 mHz, vertical blue lines) has no affect. Predicted adduct levels decay rapidly after each flash (inset); **c** 11 h after addition of chemotherapeutics, the reporter response has stabilised, and tenfold increased flash rate (every 12 s, 83 mHz) achieves detectable slow reduction of JNK reporter response, consistent with adduct prediction (inset). The reporter response gradually recovers in darkness; **d** Experiments in **c** are repeated using optoJNKi-sr, harbouring AsLOV2-V416I in as in Fig. 8, slowing relaxation tenfold (Supplementary Fig. 14). The lower flash rate (8.3 mHz) is now sufficient to maintain substantial adduct state (inset) and inhibit the pathway, as judged by the reporter readout, while reducing tenfold both photon dose and time spent on an imaging field. The latter permits the imaging of multiple samples in parallel (12 here compared with only 1 in **b**, **c**. **e** Samples, 11 h after chemotherapeutic addition, were imaged as in **d** with an extended actuation period (60 min every second hour) repeated five times. Inhibition and recovery reaches equilibrium in lit and dark phases. **f** The averaged JNK pathway reporter responses to light in **d**, **e** were fitted to exponentials with best-fit time constants shown (tau) as estimates of the reduction and recovery rates of reporter activity. Comparison of time-constants by F-test indicates significant two- to threefold differences between rates of inhibition and reactivation 20′ later ($F(1,15) = 13.46$) and of reactivation after 20′ vs. after 60′ inhibition ($F(15, 55) = 9.611$), suggesting different reactivation mechanisms depending on the duration of inhibition. The small difference between inhibition and reactivation rate at 60′ is also seen ($F(1,55) = 39.61$). ***$P < 0.0001$, **$P = 0.0023$, n.s. not significant. Source data are provided as a Source Data file. Means ± SEM ($n = 4$ wells) are shown in **b**–**e**.

Guillaume Jacquemet[42] and Stefan Jakobs[43]. VinculinTS, a gift from Martin Schwartz (Addgene plasmid # 26019; http://n2t.net/addgene:26019; RRID: Addgene_26019; ref. [44]) was used as a template for mTFP1. mTurquoise2 was generated by PCR-based mutation from a SECFP template in a plasmid also containing the Ypet template (generous gift of Michiyuki Matsuda). The JNKktr insert was obtained by PCR from pENTR-JNKKTRmRuby2, a gift from Markus Covert (Addgene plasmid # 59148; http://n2t.net/addgene:59148; RRID: Addgene_59148; ref. [22]). The AAV plasmid backbone was derived from pAAV-hSyn-3xFLAG-WPRE (RRID:Addgene_127862), generated previously[45]. The plasmids used in/built for this study are listed in Supplementary Table 3. The optoJNKi (GenBank AQY77481.1, RRID:Addgene_89744), optop38i3 (AQY77484.1, RRID:Addgene_89749) and optop38i5 inserts were developed and characterised previously[6]. The optogenetic optoNES cassette, consisting of LOV2 fused to NES21[9] was derived by PCR procedures from optoJNKi[6] as a template and linked to different fluorescent proteins as indicated in Fig. 1b and preceded by the NLS motif sequence described in Niopek et al. (ref. [9]). For cloning convenience the linker used differs from the originally described LEXY construct, as the purpose of this study was to evaluate the impact of RET on LOV2 switches in general, not specifically to manipulate the original LEXY. OptoNES spacer constructs that are used to increase the interdipole distance and reduce RET, replaced the N-terminal NLS on the fluorescent protein tag with a 3 copy NLS sequence between the fluorescent tag and the LOV2 domain, as shown (Supplementary Fig. 8). This was achieved by ligating the LOV2-NES21 cassette into the corresponding 3xNLS vectors. LOV2 constructs starting at residue 408 were obtained by PCR in a similar way. Point mutants were generated by overlapping PCR. The remaining modifications were achieved by restriction and ligation to generate the corresponding constructs. All constructs presented here for the first time (Supplementary Table 3) are available through addgene.org.

**Cell culture**. Hippocampal and cortical neuron cultures were prepared from P0 rats of either sex (mixed)[46]. Briefly, rat hippocampi and cortices were dissected separately in dissociation medium (81.8 mM Na2SO4, 30 mM K2SO4, 15.8 mM MgCl2, 0.252 mM CaCl2, 1.5 mM Hepes, 20 mM Glucose, 1 mM Kynurenic acid, pH 7.4) and digested with 50U papain, followed by 50U DNaseI treatment. Tissue were triturated with 5 ml, 1 ml, 200 μl pipette tips (about ten times each) sequentially until the cells were dissociated and homogenous. All isolation of cells and tissues from animals was performed in accordance with corresponding local, national, and European Union regulations, under approval number KEK/2015/2508 issued and renewed by the University of Turku Central Animal Laboratory. The cells were cultured in growth medium (Neurobasal$^{TM}$-A medium supplemented with 2% B-27, 1 mM glutamine and 50 U ml$^{-1}$ penicillin, 50 μg ml$^{-1}$ streptomycin) and plated in 384 well plates in the concentration of 150,000 and 250,000 cells cm$^{-2}$ for hippocampal and cortical cultures, respectively.

HEK293 cells were cultured in high glucose Dulbecco's modified eagle media supplemented with 10% (v/v) heat-inactivated foetal bovine serum, 2 mM glutamine, 50 U ml$^{-1}$ penicillin, 50 μg ml$^{-1}$ streptomycin. Cells were maintained in humidified 5% $CO_2$ atmosphere at 37 °C. The cell line was from ATCC.

**Generation of adeno-associated viruses (AAVs)**. HEK293 cells were plated in six-well plates and transfected at 50% confluency with the AAV helper plasmid, AAV-DJ replicase/capsid and AAV plasmid (Supplementary Table 3) at a ratio of 1:1:1 using the calcium phosphate co-precipitation method (see Transfection and AAV addition, below). 72 h post-transfection, supernatant of the HEK293 cells were collected and centrifuged at $500 \times g$, 10 min to remove cell debris. Cleared supernatant were aliquoted, snap-frozen in liquid nitrogen and stored at −70 °C.

**Transfection and AAV addition**. For experiments in HEK293 cells, cells were transfected with corresponding plasmids using the calcium phosphate co-precipitation method, a procedure for which has been included in the step-by-step protocol for RET-enhanced measurement of relaxation time constants of LOV2-based actuators[47]. Briefly, for one 35 mm dish of cells, a total 2.5 μg DNA was mixed with $H_2O$ and 6.25 μl 2.5 M CaCl2 at a total volume of 62.5 μl. DNA and CaCl2 were added to the same volume of 2XHebs buffer (274 mM NaCl, 10 mM KCl, 1.4 mM Na2HPO4, 15 mM D-glucose, 42 mM Hepes, pH 7.10) while swirling. After 30 min incubation under ambient conditions, the suspension of precipitate was distributed evenly over the cultures dropwise. For experiments in neurons, corresponding AAV supernatants were added in the culture and experiments were carried out 7 days after AAV addition.

**Spectral comparisons**. All FP spectra were obtained from FPbase[48], except for Dronpa off-state absorption spectra that were taken from ref. [11]. LOV2-bound FMN spectra were obtained from ref. [38].

**Image acquisition**. For imaging, neurons were plated in 384 well cell carrier plates (Perkin Elmer cat 6057302) and HEK293 cells were plated on 96-well microclear plates (Greiner cat 655090). All image data were acquired with a BD pathway 855 High-Content Analyzer (Beckton Dickinson) running Attovision V1.6, equipped with high power LED sources at 405, 445 and 623 nm, as well a white light LED source (SOLIS-405C, 445C, 623C and 3C, respectively, Thorlabs). mTurquoise2, Ypet, mScarlet and miRFP670 were excited through 438/24 nm, 500/20 nm, 555/28 nm, and 635/20 nm excitation filters, and dichroics at 458, 515, 595 nm, and a multiband dichroic for the miRFP670 channel (Chroma #84000). Images were acquired through emission filters 483/32 nm, 542/27 nm; 645/75 nm and 745/75 nm. Filters were from Semrock and Chroma. The LEDs were electronically shuttered through an optocoupler/double-inverter TTL circuit for precise exposure times and synchronisation with camera acquisition. The rest of the imaging system was controlled by Attovision software with macros according to the experimental design as described below. All quantifications were carried out on images acquired with a ×10 N.A.0.4 air objective (Olympus). Example images in Fig. 1a and Supplementary Fig. 5 were acquired with a ×40 N.A. 0.95 air objective (Olympus). Binning of 2 was used in all cases, with a resulting pixel size of 1.2 and 0.30 μm with the low and high magnification objectives respectively.

**Nuclear translocation imaging**. In Fig. 1, multiple miRFP670 images were captured to localise nuclear regions, followed by ten images each of mTurquoise2, Ypet and mScarlet at a rate of one every 10 s. The next well was then imaged and any one well was not visited again for at least 10 min to allow recovery of nuclear translocation. Exposures for miRFP670 (1 s), Ypet (0.5 s) and mScarlet (1 s) were kept constant in all cases and 438 nm exposure for mTurquoise2 and LOV2 activation was varied over the range indicated in the figures (10, 30, 100 and 300 ms, corresponding to 14–420 μmol photons.m$^{-2}$) to determine the sensitivity of the response. To ensure consistent exposure conditions in all cases, all wells were exposed to the same series of excitations regardless of whether the well expressed only one fluorophore or all three. Each well was revisited for each exposure setting (10–300 ms) and this was repeated multiple times. The data shown for mTq2, Ypet and mScarlet fusions of optoNES were obtained from an average of four visits at each setting using hippocampal neurons at 8 days in vitro ($n = 6$ replicates).

In the case of Dronpa-optoNES, the experimental comparison in Fig. 1 (and Supplementary Fig. 9) was between Dronpa-off and Dronpa-on state at distinct illumination intensities. We established the sensitivity of Dronpa to photoswitching in our system in pilot experiments. Dronpa is known to be switched to fluorescent state with exposure to 405 nm light, and to non-fluorescent 400 nm-absorbing state

(Supplementary Fig. 9, <10% maximal fluorescence[11,43]) with 488 nm light. The spontaneous relaxation is from dark-state back fluorescent state with a reported half-time of 840 min[43], therefore there will be no significant relaxation during our experiments. We found 438 nm light was able to cause Dronpa on-switching, but 470 nm, which is similarly absorbed by LOV2 (Supplementary Fig. 1), could not. Therefore 470 nm was selected for LOV2 switching in place of 438 nm so as to avoid simultaneously affecting Dronpa state. With our illumination system, we were able to switch Dronpa off most effectively with 500 nm light (~80% drop in fluorescence with 30 s exposure) so this was selected for the Dronpa off-switching phase. At 405 nm, we achieved ~20% reactivation in 0.1 s. Therefore in the experiments we used 300 s at 500 nm, equivalent to 0.231 mol m$^{-2}$ to switch Dronpa off (~10% of initial fluorescence, Supplementary Fig. 9E), and 0.5 s at 405 nm (2.3 mmol m$^{-2}$) to switch Dronpa on just prior to a 500 nm image (0.1 s, equivalent to 77 µmol m$^{-2}$) to visualise nuclear:cytoplasmic distribution. To ensure similar imaging conditions are used for quantifying nuclear:cytoplasmic ratio when Dronpa is in off or on state, only this 500 nm image was used to determine the ratio. Therefore, unlike Fig. 1, where the ratio was measured ten times during the LOV2 activation, in the Dronpa case, the ratio was only measured once at the end of each LOV2 activation series. After the LOV2 activation, Dronpa-on switching and single 500 nm image acquisition, cells were allowed to recover for 5 min. Prior to the second LOV2 activation of the cycle, with Dronpa in on-state in this case, a 500 nm image was acquired to determine the resting state ratio. This was used as a baseline ratio to quantify the response caused by each LOV2 activation train in the cycle. In all experiments three such cycles, as shown in Supplementary Fig. 9, were carried out. Average ratio changes after LOV2 activation were determined by subtracting the averaged baseline ratio. These single average LOV2 response values for Dronpa-on and Dronpa-off were used as individual replicate values, at each of four LOV2 activation intensities used, and data from six-wells were used to determine mean ± S.E.M. values. The data (six replicates for each of four photon doses) were fitted to a single site model with a common maximum response parameter. Graphpad Prism was used (see curve fitting, below) to calculate the best-fit ED$_{50}$, corresponding standard error and to apply an $F$-test to determine the statistical significance of the difference between the EC$_{50}$s. The filters used for this protocol were 405/20 excitation to switch Dronpa on, 470/40 excitation to activate LOV2 without affecting Dronpa, 500/20 excitation to switch Dronpa off, and 500/20 excitation, 515LP dichroic and 515LP emission.

**Imaging-based quantification of LOV2FRET.** In Figs. 2–6 and 8, HEK293 cells were transiently transfected with AAV plasmids. Expression from the plasmids used was driven by human synapsin I promoter in all cases. Although this is a neuron-specific promoter when DNA is delivered at low copy number as an AAV particle, we find that the promoter generates low level expression in HEK293 cells under transient transfection conditions while avoiding overexpression issues. The purpose of the RET imaging was to monitor changes in emission of a fluorophore —mTurquoise2, Ypet or mScarlet—fused to LOV2 during switching from LOV2-445 to LOV2-390 during repeated flashes of 438 nm light (at about four times a second) and relaxation back to LOV2-445 during darkness for the time period specified in the figures. As even well-optimised fluorescent proteins intrinsically exhibit a level of partial rapid photobleaching[49,50], it was important in these experiments to normalise this data to the properties of LOV2-free fluorescent protein control plasmids under identical conditions. A small initial quench was observed with unfused mTurquoise2 protein used as a control (Supplementary Fig. 10), therefore all dequenching data presented in this report are normalised to identical experiments using the corresponding unfused fluorescent protein expressing cells as a control.

**JNK pathway imaging, LOV2 activation and cell stimulation.** For experiments shown in Fig. 7, cerebrocortical cultures were infected with AAVs encoding the fluorescently tagged optoJNKi construct as specified, together with a nuclear segmentation marker fused to a different fluorescent tag, and the JNKktr reporter previously described[22] in this case fused to miRFP670 and inserted into an AAV plasmid. For mTurquoise2 and Ypet-optoJNKi, Cherry-3xNLS was used as a segmentation marker. For mScarlet-optoJNKi, Ypet-x3NLS was used. This allows capture of nuclear images for segmentation without activating the LOV2 domain. All data acquired was from neurons because the constructs, listed in Supplementary Table 3, were expressed from the neuron-specific human Synapsin1 promoter. A final 2 µg ml$^{-1}$ anisomycin was added to the neuronal culture immediately before imaging. The wells defined as light-off were sealed with aluminium sealing film (Costar) and the wells with light on were sealed with transparent sealing film (Sarstedt). An array of addressable LEDs (SMD5050/WS2812B, Adafruit), one LED above each group of 4 wells of a 384 well plate, was placed above the cell plate (Fig. 7a) to provide pulsed blue illumination centred around 465 nm. The LED array was controlled by a microcontroller synchronised via double-inverter circuit to the TTL driven camera/microscope excitation path to limit simultaneous exposure from above the plate during image acquisition and provide illumination as defined in Fig. 7b. Images were acquired for yellow (Ypet) or red (mCherry/mScarlet) and infra-red (miRFP670) channels as defined in the Imaging section above. Each well of a group was visited in turn, amounting to one visit every ~400 s. During each visit one pair of infra-red (JNKktr reporter) and red (nuclear marker) images was captured from a single field per well. For wells expressing mScarlet-

optoJNKi and Ypet-3xNLS, yellow images were taken in place of red images. The yellow or red images were used for segmentation and the nuclear:cytoplasmic ratio of miRFP670 was taken as an indicator of activation of the JNK pathway[22].

**Calculation of nuclear translocation.** All acquired image time-stacks were background-subtracted (rolling ball 50 pixels) and aligned in groups of 10. For each series of 10 time points at 10 s intervals, a maximum projection of the preceding miRFP670-3xNLS image time-series (15 images) was used to segment nuclei. These initial nuclear ROIs were eroded for the final nuclear ROIs for calculation, and expanded as a band to define a cytoplasmic ROI. For simplicity, and considering that it is neither practical nor necessary to accurately capture the entire cytoplasm of a neuronal cell under our conditions, cytoplasmic ROIs were not further processed and therefore contained some background area. This means the cytoplasmic ROIs were consistent under-estimates. Thus a visually even nucleocytoplasmic distribution (e.g. Fig. 1a, first frame) generated a nuclear:cytoplasmic ratio >1 (e.g. average 2.37 for mTq2-optoNES, Supplementary Table 1). This systematic nuclear bias as described was applied blindly to all samples and should not affect the relative changes. Most importantly it will not affect the rank photon-sensitivity of the different constructs which is orthogonal to the actual ratio values. Similarly, although spacer constructs had a more effective NLS sequence (Supplementary Fig. 8 and Supplementary movies 4–6), evidenced by over twofold higher baseline nuclear:cytoplasmic ratios (Supplementary Table 1), this should not affect the photon-sensitivity of the constructs. Mean intensities in nuclear and cytoplasmic ROIs were used for thresholding (removal of very dim and very bright cells) and remaining ROIs were used to determine nuclear:cytoplasmic ratios at each time point. At each time point ratios from all cells in a field were averaged to generate a single ratio per time point for each 438 nm exposure and each well revisit. Individual time point values for each well revisit were averaged to improve signal to noise ratio. This generated data from each microscope field as shown in Fig. 1c. Only one field was captured from each replicate. Linear regression yielded a gradient and offset value from data from each well for each 438 nm photon dose, the gradient representing the rate of change of nuclear:cytoplasmic ratio per flash of light at the given dose. Means ± SEM of these values were determined from six replicates for each construct shown in Fig. 1 and these were plotted against photon dose as shown in Fig. 1e, f. Each gradient-dose curve was fitted to a single-site model

$$\text{Response} = \text{Max} \times \text{PhotonDose}/(\text{ED}_{50} + \text{PhotonDose}), \tag{1}$$

using GraphPad Prism software to derive best-fit photon sensitivity and max response values and their standard errors, as shown in Supplementary Table 1. All scripts and data are available.

**Calculation of fluorescent protein dequenching and quenching in situ.** Quench and dequench experiments were carried out on HEK293 cells transfected with a single fluorescent construct only, as indicated. Images were acquired by repeated imaging and, for relaxation time calculations, defined periods without imaging as indicated in the figures. The pattern shown was repeated to allow data averaging and reduction of signal to noise. Image time-stacks were background-subtracted (rolling ball 200 pixels). ROIs were generated by segmentation of maximum projections of the entire image timecourse sets in each case. Intensities from all ROIs segmented in each field were averaged. Only one field was captured per well, but repeated data captured from each field was averaged, for each time point. This provided one quench/dequench curve per replicate for each condition and averages of replicates are shown. SEMs are not shown in all the main figures for clarity but are available in the source data for each figure.

Acquired dequenching curves in response to 438 nm illumination of mTq2-LOV2 constructs (Figs. 2–5, 8) were fitted to exponentials and the corresponding quench curves (for Ypet and mScarlet fusions) were fitted similarly (Fig. 6 and Supplementary Fig. 13), providing time constants and max. apparent dequench/quench (at the given intensity; lower intensity results in reduced max. apparent dequench/quench because of concurrent relaxation—see Fig. 2e). Repeated dequench/quench-relaxation curves (Figs. 3–6, 8) were fitted to a common time constant but freely fitted max. apparent dequench/quench. In the relaxation experiments, the apparent dequench or quench was plotted against the preceding dark time and this curve fitted to an exponential to derive relaxation time constants and a better estimate of apparent dequench/quench (given the concurrent relaxation). The derived constants are shown together in Table 1.

**Calculation of JNK pathway activity.** Activation of the JNK pathway reporter JNKktr[22] was quantified as the ratio of reporter signal detected in nuclear and cytoplasmic compartments. These were obtained with ImageJ scripts similar to that used for the analysis described in the section Calculation of Nuclear translocation. Multi-channel image time-stacks were background-subtracted (50 pixel rolling ball subtraction) and aligned. The locations of cell nuclei were determined by segmentation of maximum projection images of NLS-fused fluorescent proteins imaged, as described in the section JNK pathway Imaging. Nuclear and cytoplasmic ROIs were defined as described in the section Calculation of Nuclear translocation. Intensity ratios between cytoplasmic-nuclear ROI pairs were calculated, after thresholding each ROI for minimal and maximal intensities to remove very dim

cells and near-saturating cells. Raw ratio values were averaged from all cells in a field, and normalised to the first two time points of each well. Normalised field-average response curves were used to generate mean ± SEM time course curves shown in Figs. 7 and 9.

**Light calibration**. Intensities at the sample plane were determined with a power metre with microscope-stage attachment (Ex-Cite XR2100 with X-Cite XP750, LDGI), set for the relevant central wavelength.

**Measurement of dequench recovery in cell lysates**. A step-by-step protocol for RET-enhanced measurement of the relaxation time constants of LOV2-based actuators has been deposited at protocol exchange[47]. HEK293 cells were transiently transfected with CMV-driven optoNES, mTq2dC7-408LOV2-NES21, mTq2dC10-408LOV2-NES21, mTq2dC11-408LOV2-NES21 constructs using calcium phosphate precipitation. Cells were lysed 48 h post-transfection with low stringency buffer (LSB[51]) supplemented with protease inhibitors and detergent 0.5% Igepal. Lysates were cleared by centrifugation at 20,000 × g, 4 °C for 10 min. In total, 20 μl of cell lysate was transferred to a half-area 96 well μClear plate (Greiner cat 675090) and 5 μl of mineral oil (Sigma-Aldrich) was added to prevent evaporation. Before each measurement, the plate was equilibrated to either 25 or 37 °C in a plate reader (POLARstar OPTIMA, BMG) for 5–10 min. The stability of the dark state was determined by measurement after temperature equilibration. The adduct state was generated by 30 s exposure to blue LEDs (peak ~465 nm) at 2 mW cm$^{-2}$ using the 96 well trans-illuminator device described above under the section JNK pathway imaging, LOV2 activation and cell stimulation[6]. The recovery to dark state was quantified as a recovery from dequench by transferring the plate to the reader and starting the measurement protocol immediately. This protocol consisted of measurement cycles set to flying mode (3 xenon flashes per well). The excitation filter (430/10 nm) and the emission filter (475/10 nm) were used. Only 8 wells were used per measurement to minimise the cycle time (3 s per cycle). A total 20 cycles were measured, corresponding to 20 flashes over 1 min in each of three positions per well. In a pilot experiment, even 50 Hz xenon lamp flashes at a single point in a single well caused relatively slow dequenching, indicating the 0.3 Hz flashes per point required for sample measurement itself caused insignificant LOV2 re-activation. A total of eight replicate samples were measured for each construct.

Background signal from 20 μl untransfected 293 cell lysate was subtracted from the raw signal of each construct. All cycles were normalised to the first measurement of individual construct, and the data was fitted to an exponential decay with Graphpad Prism.

**Statistical analysis**. In many cases individual samples were measured repeatedly, in which case the measurements were averaged and considered a single biological replicate. In all cases, $n$ refers to distinct samples not separate measurements from the same sample. One-sided tests were not used in the statistical analyses except where indicated.

**Curve-fitting approach and statistical analysis**. Curve-fitting was carried out preliminarily in excel and subsequently using Graphpad Prism to obtain standard error estimates, in which case symmetry was assumed. Although this was necessary for the statistical analysis of curve-fit derived estimates in Fig. 4f, statistical significance evaluated by F-test (Fig. 1g, h, 9f) did not require this assumption. Formulae used were as follows:

**Curve fitting to the single-site model**. A single-site model as follows was used to determine the photon sensitivity as an ED$_{50}$

$$y = a/(1 + b/x), \quad (2)$$

where $y$ is the measured response, $x$ is the photon dose in μmol m$^{-2}$ provided, $a$ is the maximum response, and $b$ is the ED$_{50}$ or 50% effective photon dose derived.

Statistical evaluation of differences between sample groups was carried out on selected parameters (photon sensitivity) by F-test (Fig. 1g, h).

**Curve fitting of activation cycles, variable dose (first order reversible kinetics)**. For activation experiments, all datapoints were fitted to the formula

$$F = \text{constant}.\left(1 - D_{\text{limit}} + D_{\text{max}}[\text{dose}].e^{-kapp.t}\right), \quad (3)$$

where $D_{max}[\text{dose}] = D_{\text{limit}}/(1 + k_r/k_f)$, $k_{app}$, the apparent forward rate constant $= k_f + k_r$, $k_f$ the forward rate constant $=$ dose/sensitivity, and $k_r$ the reverse rate constant $= 1/r$, the relaxation time constant.

In Prism this was expressed as

$$Y = \text{Plateaumax} \times (1 - \text{Dlim} \times (1 - (1/(1 + (\text{sens}/(\text{dose} \times r)))) \times (\exp(-((1/r) + (\text{dose}/\text{sens})) \times x)))),$$
$$(4)$$

and ED$_{50}$ was calculated as sens/r.

Significant differences in best-fit parameters were determined in Table 1 with PRISM based on the manufacturer's instructions (https://www.graphpad.com/

guides/prism/7/curve-fitting/index.htm?reg_comparing_fits_with_anova.htm) using the standard errors of the best-fit parameters and degrees of freedom for each value by one-way ANOVA followed by Bonferroni's multiple comparisons test. Differences between parameters for optoNES and either spacer- optoNES or optoJNKi are shown (****$P < 0.0001$; ***$P < 0.001$; **$P < 0.01$; *$P < 0.05$; ns $P > 0.05$); df = 593, or 394 for Ypet-spacer-optoNES and mScarlet-optoJNKi (because signals using the lowest illumination intensity did not pass threshold, and second lowest passed threshold only in two of three replicates). In the case of the curve-fitting of the data in Fig. 2c, d the first of the 50 datapoints, was discarded due to an evident systematic artefact affecting most replicates, probably related to the imperfect correction to the control in this case (Supplementary Fig. 10A). No other data points were discarded in this or any other experiment.

**Estimation of dark relaxation rate from activation-relaxation cycling (Table 1 first columns)**. The entire dataset was fit to the equation:

$$F[x] = F_{\text{limit}} - A[t] \times e^{-x/\text{Tau}}, \quad (5)$$

where $x =$ the reading number (i.e. flash number), $F[x] =$ the fluorescence measured (arbitrary units) at reading $x$, $F_{limit} =$ the fitted fluorescence limit as exposure (# flashes) $\rightarrow \infty$, $A[t] =$ the fitted recovery from (de)quenching after the defined dark period t, and Tau = the shared activation rate constant in units of flashes, $=$k$_{app}$/photon dose per flash

In Prism, each of the 12 cycles occupies a separate data column with unique A[t] but shared Tau (see source data). In Excel the data can be fitted as a single column with row-specific $A[t]$ values but this does not directly provide standard errors of the fitted constants. In Figs. 3e, 6c, 7, the apparent activation rate or forward rate constant is shown as $k_{app}$ in μmol m$^{-2}$ at the photon dose per flash used.

The $A_t$ values obtained above for each dark time were plotted against dark time $t$ and fitted to the exponential

$$A[t] = A_0 \times \left(1 - e^{-t/r}\right), \quad (6)$$

where, $t =$ the dark time, $A[t] =$ the best-fit values of the recovery from dequenching or quenching after the dark time $t$, from the curve fit above, $A_0 =$ the fitted maximal increase in fluorescence as dark time $\rightarrow \infty$, and $r =$ the relaxation time constant, shared across cycles with the same construct.

This curve is shown in Figs. 3d, 4d, 5c, 6d, g and 8c, where the dequench or quench maxima at the photon doses uses were calculated from the maximal change in fluorescence as a fraction of the maximal fluorescence. Thus max. dequench max. at the photon dose per flash specified in the tables = A0/(1 + A0) as a %.

**Quantification of dequench limit, ED$_{50}$ and sensitivity**. The above fit was carried out to help visualise the quench and dequench recovery under the conditions used. To acquire more fundamental properties the equation below was fitted to all datasets. One exception was the mTq2-optoNES v mTq2-spacer-optoNES dataset (Fig. 3). In this case the response of the spacer construct was low and the fits to individual datasets generated a poor fit. Therefore the averages ($n = 5–6$) were used instead for fitting to the equation below, which produced a much better fit.

Dequench limit, sensitivity and ED$_{50}$ was obtained by fitting all datapoints to the following equation:

$$Y = \text{Offset} \times \left(1 - D_{\text{limit}} + \left(D_{\text{max}}[\text{dose}] \times \left(1 - \left[(1 - e^{-kr.t}).e^{-kapp.x/4}\right]\right)\right)\right), \quad (7)$$

where $D_{max}[\text{dose}] = D_{\text{limit}}/(1 + k_r/k_f)$; $k_{app} =$ the apparent forward rate constant $= k_f + k_r$, $k_f =$ the forward rate constant $=$ dose/sensitivity, 56/sensitivity in μmol m$^{-2}$ in this case, $k_r =$ the reverse rate constant $= 1/r$, the relaxation time constant.

In Prism this equation was expressed as:

$$Y = \text{Offset} \times (1 - \text{Dlim} \times (1 - ((1/(1 + (\text{sens}/(56 \times r)))) \times ((\exp(-\text{darktime}/r) - 1) \times \exp(-x \times ((1/r + 56/\text{sens})/4)))))), \quad (8)$$

where the fluorophore acted as an acceptor and was quenched, the following equation was used instead, providing values for quench limit, sensitivity and ED$_{50}$

$$Y[x] = \text{Offset} \times \left[1 - Q_{max}[\text{dose}] \times \left[1 - \left(1 - e^{-kr.t/r}\right) \times \left(1 - e^{-kapp.x/4}\right)\right]\right], \quad (9)$$

where $Q_{max}[\text{dose}] = Q_{\text{limit}}/(1 + k_r/k_f)$; $k_{app} =$ the apparent forward rate constant $=$ k$_f$ + k$_r$, $k_f =$ the forward rate constant $=$ dose/sensitivity, 560/sensitivity in μmol m$^{-2}$ in this case, and $k_r =$ the reverse rate constant $= 1/r$, the relaxation time constant.

In Prism this equation was expressed as:

$$Y = \text{Offset} \times (1 - ((Q_{\text{limit}}/(1 + (\text{sens}/(560 \times r)))) \times (1 - (1 - \exp(-\text{darktime}/r)) \times \exp(-x \times ((1/r + 560/\text{sens})/4))))), \quad (9)$$

In both cases, ED$_{50}$ was calculated from sensitivity/4r.

Dark periods were 1–180 s as shown. The first dequench was set arbitrarily as 600 s because a minimum of 10 min was left between well revisits and this is far above all time constants measured.

**Analysis of JNKktr reductions and recoveries**. Data from all replicates ($n = 4$) and all reduction phases or recovery phases (4–5 repeats per replicate), during blue-light illumination or blue-light darkness were averaged. The averaged curves were fitted to exponentials. The impact of replacing the dark-phase recovery time constant with an exponential decay, corresponding to the slow relaxation of the optoJNKi-sr inhibitor, was considered. Thus in the parameter $e^{-(kf+kr).t}$, the uninhibited forward rate constant $k_f$ was replaced with $(1-A.e^{-r.t}).k_f$ where $A$ represents the maximal fractional reduction of $k_f$ by the optogenetic inhibitor, and $r$ the rate constant at which optogenetic inhibitor relaxes, ~2 min in this case. This had minimal effect (~10%) on fitted time constants, even on the fastest kinetics. Therefore the slowed relaxation of optoJNKi-sr cannot explain the slower recovery of JNKktr signal after 20' inhibition compared with after 60' inhibition (Fig. 9d, e). This decaying time constant was not routinely included, to limit the number of unknowns in the curve-fit formulae. To compare JNKktr reduction and increase in the same curve fit for each experiment, spans were set as a shared variable, and the fitted constants were compared by $F$-test, the results of which are cited in the figure and in the results text.

**Simulation of adduct state fraction**. It is possible to simulate the expected proportion of actuator in adduct fraction over time, on the basis of the optogenetic actuator sensitivities and relaxation times, together with specific input of blue light over time. Sensitivities used were based on measurements with 438 nm light, because AsLOV2 absorbance at 438 nm (used in microscopy) and 465 nm (from the parallel LED trans-illuminator device) are very similar. An excel sheet generating the simulation with the desired time step resolution is provided as a Supplementary file.

**Reporting summary**. Further information on research design is available in the Nature Research Reporting Summary linked to this article.

## Data availability
Source data are provided with this paper. Figure data are in source data files. Raw imaging data, together with associated metadata files, for Figs. 1–9 are available at https://doi.org/10.5281/zenodo.3882572[52]. Scripts used for controlling the image acquisition are within this metadata. All data in this manuscript are available from the corresponding author upon request. All plasmids used in this manuscript were new plasmids and deposited in addgene under IDs 159941-159976. They are described in detail in Supplementary Table 3. These plasmids were generated in part with the help of previously generated plasmids (including addgene IDs 89744, 89749, 127862) and plasmids provided by other labs (including addgene IDs 26019, 59148). The FBbase database was also used for this work (https://www.fpbase.org/). Any other relevant data are available from the authors upon reasonable request. Source data are provided with this paper.

## Code availability
Scripts for analysing the optoNES translocation and in situ photocycling, outputting the source data, are in Supplementary files. The excel calculator for predicting adduct state over time is in the Supplementary files.

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

## Acknowledgements

This work was supported by the Magnus Ehrnrooths Foundation (M.J.C.), the National Cancer Institute Grant R01CA200417 (M.J.C. and L.L.L.), the European Union Erasmus + programme (F.M.K.), and 7th Framework Programme Initial Training Networks FP7-PEOPLE-2013-ITN Project Number 608346 Project 'Brain Imaging Return To Health' r'Birth (A.P., L.L.G.), the Academy of Finland (mobility projects 309736 and 324581 to M.J.C.), the DAAD with funds from the German Federal Ministry of Education and Research (57348387 and 57458932 to F.F.), and by access to the facilities of the Turku Screening Unit, a member of the Biocentre Finland Drug Discovery and Chemical Biology Network. We thank David Kelly (Beckton Dickinson) for advice on Pathway 855 instrument maintenance, Michiyuki Matsuda (Kyoto University), Stefan Jakobs (Max Planck Institute for Biophysical Chemistry, Göttingen) and Guillaume Jacquemet (Åbo Academy University) for generously providing fluorescent protein templates, Peter Martinsson (University of Eastern Finland) for generating the Dronpa coding sequence, Elena Tcarenkova (University of Turku) for helpful discussion and James Conway (University of Turku) for critical reading of the manuscript.

## Author contributions

L.L.L. prepared samples and reagents, and designed and carried out the experiments, analysed the data and performed statistical analyses. F.M.K. prepared samples and reagents, and performed experiments and analysed the data. L.L.G. participated in the preparation of initial optoNES constructs. A.P. participated in the preparation of AAV vector systems. F.F. assisted and advised on application of the AAV system. M.J.C. designed the project and experiments, prepared reagents, carried out experiments, analysed data, and prepared the manuscript. All authors assisted in drafting the manuscript.

## Competing interests

The authors have no competing interest
