## [Peer Review File · Nature Communications]

Reviewers' Comments:

Reviewer #1:

Remarks to the Author:

This is an interesting paper by Li et al. I recommend that the manuscript be reconsidered after revisions.

Specific comments:

1. In the first part of the paper, the authors describe the ability to increase the sensitivity of the photoswitch by using FRET. However, the effect is fairly modest (2-fold relative to a system with no FRET?) It is not clear why a system with decreased sensitivity (due to FRET) would be useful and so the indicated "range" may be a bit misleading.
2. The authors should check their 9 nm estimate of length for a 24 amino acid spacer. 9 nm seems like it would be closer to the "contour length" of a 24 amino acid polypeptide. I would think that the RMS (root-mean-squared) end-to-end distance for a 24 amino acid random coil would be smaller. The authors should verify the numbers and consider the implications if any for their analysis. (Is a 24 amino acid spacer "long enough"?)
3. The use of FRET to characterize relaxation times is neat and one of the strongest parts of this manuscript.
4. The finding that relaxation times are considerably faster in cells than in vitro is an important one. Do the authors have a hypothesis to explain this result? Could some experiments help them choose among competing hypotheses? It would be unfair to ask that the scope of a single paper be broadened too much, but this is an important finding and deserves some scrutiny/analysis.
5. The application of the FRET approach to other optogenetic actuators is a good demonstration of their technique.
6. Characterization of the opto-JNKi-sr relaxation time is good, but some minor rephrasing of text towards the end should be considered. While the authors have characterized the relaxation time in cells, the effect of the V416 mutation on relaxation times has been reported previously and is the primary reason for the improved throughput.
7. There are some minor spelling mistakes (e.g., "timecouse") which can be easily corrected.

Overall, this is a thorough study and I recommend publication after revisions.

Reviewer #2:

Remarks to the Author:

This manuscript describes the analysis of the effect of fluorescent proteins (FP) fused to LOV2 optogenetic probes. Specifically, the authors provide clear evidence that under appropriate conditions, resonance energy transfer can occur between the the FP and the LOV2 domain (or vice versa depending on the FP). This energy transfer can serve to sensitize or desensitize LOV2 to activating light. Furthermore, the related quenching of the FP can be used to assess the state of activation of the LOV2 domain, potentially allowing optimization of illumination protocols to limit the impact of blue light in optogenetic experiments.

While the above represents the strong parts of this manuscript there are significant weaknesses.

- 1) First, the manuscript is very long does not convey the impression of a carefully assembled manuscript. The figures are not well curated, they are not well assembled, they contain typos, and the text is too long and frequently lacks clarity. Regarding curation, two clear examples, Figure 1 is a re-representation of FPBase information. Figure 2 Ei and Fi are crude experiments and are

superseded by Figure 2 Eii and Fii. The first two do not provide any benefit and should not be in the manuscript. This would allow deletion of lines 166-187, the net result being a gain in clarity. There are a number of problems in the text, perhaps most noticeably the fact that the discussion ends with a list. It is the job of the authors, not the reviewers, to better curate and present their data. Nevertheless, some suggestions for improving the figures and text are appended below.

2) There are issues with the underlying premise of the manuscript and the interpretations. The authors suggest that the spectral behavior of the FP directly reports on the behavior of the optogenetic probe. This is only true in cases in which the optogenetic effector exchanges as fast as the LOV2 domain returns to the dark state. There may be cases in which the off rate is slower in which case the state of the FP is not indicative of the instantaneous state of the probe. For example, the off-rate of Lov-ipaA is far slower than the recovery rate of the LOV domain (Lungu, 2012). This is an extreme case, but similar effects could be seen with other switches and the authors appear to assume that this is not the case. This over interpretation/limitation must be corrected.

3) Second, again to the premise of the manuscript, the authors state on line 37, " However, long-term optogenetic modulation of pathways can be problematic as blue light can itself perturb cells." While this is true, it is not typically the case. First the authors typically use 438 nm light, whereas 488 nm light is also effective at activating the LOV2 domain and less toxic. Second, in the vast majority of experiments, the state of the system as a consequence of optogenetic probe activation will be assayed by measuring the state of the system with FPs. In practice, the amount of light required for activating LOV2 is often small relative to those necessary for measuring the response. This is not to say that managing the "light budget" isn't important, but the primary value of this approach is the window it can provide to the behavior of the probe.

In summary, this study presents an interesting approach that could be of utility for the field. This work could be suitable for publication in Nature Communications if it is subject to an extremely thorough revision that clearly presents the findings and its limitations in a manner that is comprehensible to a general audience.

Text issues

Line 43-45 " The greatest value of optogenetics is its unique potential to impose or modulate complex time-encoded inputs to any desired target in living cells, and to establish long-term manipulation and mimicry of a range of periodic and pulsatile biological phenomena"
The authors state this as a fact and many in the field do not view this as "The greatest value of optogenetics"

line 70 " bidirectional resonance energy transfer (RET) between the fluorescent proteins and the LOV2-bound flavin mononucleotide (FMN)." RET can occur in one direction with a given FP and the LOV2 domain. It is not bidirectional.

lines 197-199 are superfluous

line 348-51 "48 To determine whether it is generally possible to monitor activation sensitivities and relaxation rates of different LOV2-based constructs in situ, we investigated whether these parameters could be even known whether the activation sensitivities and relaxation times in intact cells are common for all LOV2-based constructs or whether these parameters vary widely among different constructs. "
Not clear.

Line 464. The authors state, "However their [mutating V416, I427 and F434] effects on LOV2 switching have not previously been directly measured in situ."
V416I has been studied in vivo PMID 22388287

Line 542-4 "However, implementation in cell biology and related fields has been surprisingly limited to date, with a high proportion of publications currently in the form of reviews."

This is quite dismissive of some very important studies from many labs and particularly odd given that a search for "Courtney-T and optogenetics" on PubMed reveals one research paper and two reviews.

562-4 " However, inhibitors able to block endogenous signalling processes are often more valuable tools for investigation of physiological mechanisms than approaches that impose an overexpressed synthetic activity."

Both types of studies can have value, it depends on how they are deployed.

Figure issues and suggestions

Figure 1

- this is at best a supplemental figure. It does not contain any newly acquired data presented here.
- epsilon should be defined in the figure legend and associated with a particular fluorophore
- emission and excitation spectra should be distinctly indicated (e.g solid vs dash lines)
- shade overlap region between FP and LOV2
- the -> marker should be replaced with an arrow
- 1C, LOV2 is labeled norm, 1G excitation is misspelled, 1E Lov2 absorption is used instead of excitation
- perhaps align spectra in which LOV2 is the emitter

Figure 2

- The images in fig. 2a are too small and blurry -
- figure Ei and Fi - the co-expression of the probes - are superfluous and add unnecessary complications
- the axes are labeled in an overly complex manner. Use the figure legends to explain the details.
- the boxes around each graph are unnecessary

Figure 3

- the wavelength of light can be stated in the legend. The relevant parameter on the x-axis is the number of flashes

Figure 4

- The reading number in figure 4 is a poor choice for the x-axis. the key aspect of these experiments is that there is an increasing delay between trials of each 20 readings. The x-axis should include this feature, by adding gaps of increasing size between each set of measurements. Alternatively, simply put 4C,D as supplementary material and focus the reader on 4E (4D and 4E are essentially redundant)
- Also 4B and C should be on the same graph. If they are on different graphs, they should be the same width at a minimum.
- 4D fluorescence is spelled incorrectly
- the boxes around each graph are unnecessary

Figure 5

- The reading number in figure 5 is a poor choice for the x-axis. the key aspect of these experiments is that there is an increasing delay between trials of each 20 readings. The x-axis should include this feature, by adding gaps of increasing size between each set of measurements.

Figure 6

The results in figure 6 A-C are mostly redundant with the table.

We thank both reviewers for their very helpful criticism and detailed suggestions, which we believe helped us to improve the manuscript. Our point-by-point responses, detailing the changes we have made, are below.

Reviewers' comments:

Reviewer #1 (Remarks to the Author):

This is an interesting paper by Li et al. I recommend that the manuscript be reconsidered after revisions.

Specific comments:

1. In the first part of the paper, the authors describe the ability to increase the sensitivity of the photoswitch by using FRET. However, the effect is fairly modest (2-fold relative to a system with no FRET?) It is not clear why a system with decreased sensitivity (due to FRET) would be useful and so the indicated “range” may be a bit misleading.

We agree that we only demonstrated a 2-fold sensitization compared with a reduced FRET (“spacer”) variant. Nevertheless we also showed this represents a 7-fold sensitization when quenching red/yellow proteins are replaced with cyan. Considering how frequently red proteins are used to tag blue-light switchable proteins (e.g. PMID 29379139, 29446199, 27427858, 26853913, 26646180, 25019686, 23601651, 19693014, 25233328, 22847441, 21037589), we think it is still important to indicate both the 2-fold and 7-fold improvements (hence the 7-fold range). We have now tried to clarify the limited circumstances when that a 7-fold improvement might be possible (bold text in lines 56, 121-122, 363-364, 439-440).

2. The authors should check their 9 nm estimate of length for a 24 amino acid spacer. 9 nm seems like it would be closer to the “contour length” of a 24 amino acid polypeptide. I would think that the RMS (root-mean-squared) end-to-end distance for a 24 amino acid random coil would be smaller. The authors should verify the numbers and consider the implications if any for their analysis. (Is a 24 amino acid spacer “long enough”)?

We thank the reviewer for pointing out our error. We replaced the estimate with one for RMS distance and mentioned the following arguments in the text. A random coil would be $0.38 \times \sqrt{24} = 1.9\text{nm}$, which is quite short. However i) assuming a generous initial fluorophore distance $\sim 5\text{nm}$, adding $1.9\text{nm} = 6.9\text{nm}$ still increases $(R/R_0)^6$ by a factor of 7 and a substantial impact on energy transfer; ii) considering that the spacer includes excess positive charges to promote a more extended conformation, the RMS value is likely to be an underestimate. This point has been addressed in lines 113-116 (bold text).

3. The use of FRET to characterize relaxation times is neat and one of the strongest parts of this manuscript.

We appreciate this point and bore it in mind while greatly shortening the manuscript text to conform to journal requirements.

4. The finding that relaxation times are considerably faster in cells than in vitro is an important one. Do the authors have a hypothesis to explain this result? Could some experiments help them choose among competing hypotheses? It would be unfair to ask that the scope of a single paper be broadened too much, but this is an important finding and deserves some scrutiny/analysis.

We greatly appreciate this question. Like the reviewer, we initially expected this somewhat out of scope and perhaps intractable. But we added a new experiment and statistical comparison which simplifies and resolves this issue (Fig. 4F and section with subtitled *Determinants of relaxation rate*) We modified references to changes in relaxation rate to reflect these new results. We also demonstrated the applicability of a cell-free relaxation measurement to constructs identical to those used in situ, without any need for typical recombinant purification procedures.

Briefly, the relaxation time constant of ~80s reported for isolated LOV2Ja at 25°C is often assumed to apply to LOV2-based actuators in living cells. Temperature and Jalpha mutation are known to influence relaxation rate. But we thought the discrepancy, at 37°C and with the same LOV2Jalpha sequence, between optoJNKi relaxation in this manuscript in intact cells at ~13-16s and by cell-free absorbance spectroscopy in a previous study at 24s, indicated an impact of the intracellular environment (perhaps crowding or chaperone action).

However, the FRET optimization experiment (Fig. 4, previously Fig. 5) showed that modifying the linker N-terminal to LOV2 caused up to a 3-fold change in relaxation rate, again at 37°C and with the same LOV2Jalpha sequence. Our purified vs in cell optoJNKi comparison above was invalid because the purified optoJNKi used was a bacterial recombinant His-tagged form of optoJNKi in accordance with commonly used procedures but our in-cell experiments used a fluorescent protein tag with a distinct linker based on cloning sites, also in accordance with commonly used procedures. The two have quite different linker sequences.

Considering optogenetic actuators used typically bear fluorescent tags rather than His tags, we realized it was essential to acquire cell-free data from sequences used in mammalian systems and not those designed for optimal recombinant expression and purification.

We selected sequences from Fig. 4 with the most different relaxation rates, transferred them to CMV-driven vectors to generate sufficient protein in HEK293 cell lysates. These are diluted ~1000 times compared with intact cells so we can assume an impact of the intracellular environment is greatly reduced. We used these lysates to monitor relaxation time. The difference between relaxation at 25°C and 37°C is clear but there was no significant difference between relaxation at 37 in lysates and in cells. There remains a small non-significant difference, but this could reflect slight cooling when transferring the plate between the plate reader and the trans-illumination device.

The conclusion is that, at least for these constructs, relaxation is not influenced by intracellular environment per se. Instead, the relaxation rate is far more sensitive to flanking sequences than previously appreciated.

Thus N-terminal linker sequences typically used to tag constructs for mammalian expression themselves can have substantial impact on relaxation rate. It means any modification of an existing design should be checked for modified relaxation rate. But we also show that this is not difficult because relaxation rates (though not activation) can be determined in a plate reader, as a complement to the more automated *in situ* measurement under the microscope.

The new text is in lines 210-218 and 416-423.

5. The application of the FRET approach to other optogenetic actuators is a good demonstration of their technique.

We appreciate this point and bore it in mind while shortening the manuscript text.

6. Characterization of the opto-JNKi-sr relaxation time is good, but some minor rephrasing of text towards the end should be considered. While the authors have characterized the relaxation time in cells, the effect of the V416 mutation on relaxation times has been reported previously and is the primary reason for the improved throughput.

We agree the V416 mutation was previous described and even used in cells (lines 292-296, 431-433 in bold). The modified text clarifies how it was the knowledge of *in situ* relaxation parameters and target response time that together informed the guided our use of this mutant and that it was a previously described mutant. The reason it is an improvement is because the *in situ* relaxation time, revealed in the paper as much shorter than expected from purified LOV2, remains shorter than target response times (lines 433-435). We think the improvement also depended on the *in situ* method, because it demonstrated that the modification would not significantly compromise time resolution; this is how we tried to present the optoJNKi-sr.

7. There are some minor spelling mistakes (e.g., “timecouse”) which can be easily corrected.

There were a lot of typos that escaped our notice in the earlier version. We have now completely re-written the text and checked carefully to eliminate spelling mistakes, as well as revising figures according to reviewers’ comments also correcting typos there.

Overall, this is a thorough study and I recommend publication after revisions.

Reviewer #2 (Remarks to the Author):

This manuscript describes the analysis of the effect of fluorescent proteins (FP) fused to LOV2 optogenetic probes. Specifically, the authors provide clear evidence that under appropriate conditions, resonance energy transfer can occur between the the FP and the LOV2 domain (or vice versa depending on the FP). This energy transfer can serve to sensitize or desensitize LOV2 to activating light. Furthermore, the related quenching of the FP can be used to assess the state of activation of the LOV2 domain, potentially allowing optimization of illumination protocols to limit the impact of blue light in optogenetic experiments.

While the above represents the strong parts of this manuscript there are significant weaknesses.

1) First, the manuscript is very long does not convey the impression of a carefully assembled manuscript. The figures are not well curated, they are not well assembled, they contain typos, and the text is too long and frequently lacks clarity. Regarding curation, two clear examples, Figure 1 is a re-representation of FPbase information. Figure 2 Ei and Fi are crude experiments and are superseded by Figure 2 Eii and Fii. The first two do not provide any benefit and should not be in the manuscript. This would allow deletion of lines 166-187, the net result being a gain in clarity. There are a number of problems in the text, perhaps most noticeably the fact that the discussion ends with a list. It is the job of the authors, not the reviewers, to better curate and present their data. Nevertheless, some suggestions for improving the figures and text are appended below.

Although we tried to assemble the manuscript very carefully, we fully accept the reviewer's opinion that it does not convey this impression. We agree the initial submission was far too long. It was intended to provide all information up-front for review, but we apologise this increased the burden on the reviewers. The text has been thoroughly revised with the help of the reviewers' inputs and body text word count is less than halved. We hope to have caught all typos this time. We greatly appreciate the specific and detailed suggestions of the reviewers for improving the manuscript.

In almost all cases the revised version incorporates changes in the parts of the main text and figures highlighted by the reviewers. There are a couple of cases where we felt it was important to retain some aspect and have tried to justify these cases below.

We originally included Fig. 1 in the main figs, even though it used values from the Gardner lab and FPbase, to help illustrate the likelihood of FRET with LOV2 to the non-specialist reader. Thus we have kept it as a supplementary figure (new Fig. S1) rather than remove it completely.

We also hope it is acceptable to place the original Fig. 2Ei, Fi in supplementary data as they are genuine observations from what was intended to be a refined approach to avoid problems associated with cellular heterogeneity. It does show how striking an impact FRET does have on function of LOV2 constructs under the conditions specified. We agree there are complexities involved so we briefly explain in the main text that, for this reason, the constructs should be evaluated singly (now E and F of Fig. 1). We prefer to offer readers this chance to see the complexities that arise from co-expression and we think the way we present it now does not cause any confusion.

2) There are issues with the underlying premise of the manuscript and the interpretations. The authors suggest that the spectral behavior of the FP directly reports on the behavior of the optogenetic probe. This is only true in cases in which the optogenetic effector exchanges as fast as the LOV2 domain returns to the dark state. There may be cases in which the off rate is slower in which case the state of the FP is not indicative of the instantaneous state of the probe. For example, the off-rate of Lov-*ipaA* is far slower than the recovery rate of the LOV domain (Lungu, 2012). This is an extreme case, but similar effects could be seen with other switches and the authors appear to assume that this is not the case. This over interpretation/limitation must be corrected.

We agree the spectral measurement does not directly indicate the biochemical actuation by the construct, and did not mean to suggest this. It only informs the state of the generic optical switch (adduct or not), not the state of the downstream actuation machinery and its interaction with downstream targets in the cellular environment. The latter are actuator, target and context-specific. There can be no generic solution for monitoring case-specific properties, the actuation step nevertheless still requires the generic optical switching event and it is this that can be addressed *in situ*. Potentially, knowledge of the targets of specific actuators could be built onto the model used in the switch state calculator included as a supplementary file.

Regardless, when evaluating new constructs in cells, or implementing previously designed constructs in a new system (or new lab), we found it is very useful to know whether the photon delivery train has actually switched the protein state or not, and for how long that state remains switched.

We specifically addressed this limitation in lines 386-388 (bold).

In addition, we specified in line 269-272 (bold) that the actual actuation will last longer than the switched state and that the amount of switched state will be an underestimate of the actuator engaged with target.

Moreover, to avoid risks of over-interpretation by readers we carefully revised the text to ensure we use the term “switching” to refer to the change of FMN state that can be detected by RET (Figs. 2-6, 8), and “actuation” (Figs., 1, 7, 9), which depends also on the downstream machinery. The term “actuator” is still used in conjunction with “switching” because the actuator contains the switch (and downstream machinery).

3) Second, again to the premise of the manuscript, the authors state on line 37, " However, long-term optogenetic modulation of pathways can be problematic as blue light can itself perturb cells." While this is true, it is not typically the case. First the authors typically use 438 nm light, whereas 488 nm light is also effective at activating the LOV2 domain and less toxic.

We agree that 438nm light is more toxic than 488nm light, though LOV2 absorbance at 488nm is about 2.5 times less than at 438nm. We used 438nm because quantification of cyan protein state in the presence of other proteins (specifically the yellow protein) required the use of 438nm (to avoid exciting the yellow protein or for more sensitive recording of dequenching of the cyan protein with short exposure times).

And, for actuation, we fully agree that 488nm (using ~2.5 more photons) could be used for actuation in place of 438nm.

However, we do not completely agree that phototoxicity “is not typically the case”. Perhaps we have different cell types and time frames in mind. Neurons and stem cells may be particularly sensitive and in these cases time frames can extend from hours to days or weeks. We now clarified in the manuscript that our concerns about long-term effects refers to hours and days – line 40 (bold).

Thus for long-term experiments (e.g. in the range of hours and days) we think that there is evidence that cellular perturbation remains a concern. One paper that we cited – Duke et al., 2019 – showed detectable changes in gene expression at 8 hours, induced by extremely low intensity light (time-averaged illumination of only 0.021mW/cm² i.e. 0.84mW/cm² at 2.5% duty cycle) from 470nm LEDs. This illumination rate is well below that commonly used in optogenetic experiments and it seems unlikely that replacing 470nm (LED) with sufficient light at 488nm (i.e. 2.5 x higher intensity required) would eliminate blue light perturbations.

Moreover, trans-illumination and multiwell approaches to optogenetics typically use 470nm LEDs. This is largely because they are easy to control, inexpensive and, most important of all, arrayable at 9mm pitch or smaller, unlike 488nm sources (e.g. PMID 23601651, 25708714, 26646180, 26699507, 23877069, 24180414, 28497795, 28893998, 29379139, 30333716; we were not able to find any case of 488nm trans-illumination systems used for independent well control in multi-well optogenetics).

For these reasons, we therefore think it is fair to state in the introduction (before specific blue wavelengths such as 438nm, 470nm or 488nm are mentioned) that “long-term” modulation can be problematic because “blue light can perturb cells”, so we have kept this statement in the introduction (lines 39-40, bold) and added the reference to Duke et al, 2019 and Losi et al., 2018 in support.

Second, in the vast majority of experiments, the state of the system as a consequence of optogenetic probe activation will be assayed by measuring the state of the system with FPs.

We enthusiastically agree that FP-based assays are extremely powerful (even over a several week timescale). However, we believe that optogenetic actuators have potential for application to a much broader range of assays including those that do not involve FPs and microscopy.

In cases when the state of the system can be determined in real time with FPs – for example the translocation of optoNES or LEXY - our approach is not necessary. But this restricts usage to applications for which convenient FP-based readouts work as intended in the biological system in question. Fortunately, there are many such assays available. Equally, there are other cellular functions for which these do not exist. We state in the text that cellular optogenetics has typically been constrained to these former cases – lines 342-344 (bold).

Our method aims to support also applications that do not involve FP-based microscopy, once FP-based measurements reveal the activation and relaxation parameters that are required for switching.

Numerous applications of channelrhodopsins, for example, do not involve any measurement of FPs, because the readout is at a functional level. Similarly, genetic and chemical perturbation experiments do not always require FP use. Optogenetics has a potential be a flexible option alongside genetic and chemical perturbation approaches.

Thus we hoped to show how the parameters, once acquired on any fluorescence microscope, provides sufficient information to design illumination trains to optically switch an actuator to a chosen extent, even when an on-line FP microscopy validation assay is not feasible or available. Multiwell trans-illuminator systems (Fig. 7) can then achieve parallel and well-specific actuation of entire plates without use of a microscope (LOV2 absorption at 470nm being the same as 438nm), for any alternative application. We were obliged to use FP-based assays to most convincingly demonstrate the parameters acquired by LOV2 FRET could be used this way, We accept this is potentially confusing because, as with optoNES, in all cases where we have a convenient assay we could have relied on trial and error instead of modelling the actuation and using the recorded activation/relaxation parameters.

We have now tried to clarify this point better in the revised text, including in lines 427-429 (bold).

In practice, the amount of light required for activating LOV2 is often small relative to those necessary for measuring the response. This is not to say that managing the "light budget" isn't important, but the primary value of this approach is the window it can provide to the behavior of the probe.

Based on our data, we do not fully agree with the first sentence. We can make the direct comparison with the sensitized mTq2-optoNES construct. $14\mu\text{mol.m}^{-2}$ light was sufficient for one data-point quantifying nuclear-cytoplasmic ratio (Fig. 1) whereas $\sim 8\mu\text{mol.m}^{-2}$ should be delivered continuously at $\sim 4\text{Hz}$ to achieve half-maximal switching (Fig. 2E, J). Moreover, Fig. 1D shows $1400\mu\text{mol.m}^{-2}$ only achieves 9% actuation over 100 seconds. Therefore, we can collect many response data-points even at 438nm without much actuation.

Clearly, the situation depends on the experimental design. Long relaxation time mutants increase actuation during measurement, whereas use of infrared probes or non-optical readouts would eliminate measurement-associated blue-light altogether (Fig. 7, 9). Regardless, we propose that it would be preferable if no light at all was required for measuring the response, given that gene expression has been shown to be so easily modified (ref. 20, Duke et al., 2020). It was our aim to at least provide such an option, though we fully recognize FP reporters are extremely useful tools.

In summary, this study presents an interesting approach that could be of utility for the field. This work could be suitable for publication in Nature Communications if it is subject to an extremely thorough revision that clearly presents the findings and its limitations in a manner that is comprehensible to a general audience.

Text issues

Line 43-45 " The greatest value of optogenetics is its unique potential to impose or modulate complex time-encoded inputs to any desired target in living cells, and to establish long-term manipulation and mimicry of a range of periodic and pulsatile biological phenomena"

The authors state this as a fact and many in the field do not view this as "The greatest value of optogenetics"

We thank the reviewer for pointing this out. We have removed the term "greatest value" and added an additional recent reference to illustrate the concept we refer to (lines 41-43, bold).

line 70 " bidirectional resonance energy transfer (RET) between the fluorescent proteins and the LOV2-bound flavin mononucleotide (FMN)." RET can occur in one direction with a given FP and the LOV2 domain. It is not bidirectional.

We had used the term to indicate that RET can occur from an FP to LOV2 or vice versa, depending on the FP species. We now removed the word bidirectional from the main text to avoid possible confusion. In the specific case of homoFRET (as may occur with the TFP1 construct) we still include the term bidirectional in the supplementary information.

lines 197-199 are superfluous

We have removed them.

line 348-51 "48 To determine whether it is generally possible to monitor activation sensitivities and relaxation rates of different LOV2-based constructs in situ, we investigated whether these parameters could be even known whether the activation sensitivities and relaxation times in intact cells are common for all LOV2-based constructs or whether these parameters vary widely among different constructs."

Not clear.

This was a copy-paste error we missed during manual re-pagination, but we now provided a briefer replacement.

Line 464. The authors state, "However their [mutating V416, I427 and F434] effects on LOV2 switching have not previously been directly measured in situ."

V416I has been studied in vivo PMID 22388287

We agree V416I was used in Strickland et al., 2012 to show differences in rate of recruitment and dissociation to targets (or spots), and our earlier reference to it was somewhat unclear. We now cite Zoltowski 2009 and Zayner 2013 for the cell-free measurements and Strickland 2012 and Wang 2016 for in cell quantification of kinetics within specific actuators harbouring this mutant.

We had meant that, previously, the switching (i.e. between lit and dark states) of LOV2-V416I was not directly measured *in situ*. As the reviewer has stated (reviewer's point 2, above), this is not the same as actuation because this also depends on off-rates of actuator-target interaction. Ultimately the specific cellular actuation is the most important end-point, but our method determines only the basic switching parameters that are required for whatever actuation is coupled to it.

We have now cited Strickland, 2012 and Wang 2016 in lines 295, 344, 346, and 433.

Line 542-4 "However, implementation in cell biology and related fields has been surprisingly limited to date, with a high proportion of publications currently in the form of reviews." This is quite dismissive of some very important studies from many labs and particularly odd given that a search for "Courtney-T and optogenetics" on PubMed reveals one research paper and two reviews.

We apologise that our original wording could have been misunderstood in this way. We greatly appreciate the impressive studies that have been published from numerous labs. We had no intention to indicate otherwise and are not referring to our own work. We simply meant that the approach is not as commonplace as use of GFP or fluorescence microscopy. We believe the method is so powerful that it should be much more widely used. In particular, as many labs have shown, it does not even constrain the user to microscope-based approaches. I hope we can make this point without appearing dismissive.

We have rephrased our point – lines 339-340, in bold – in the hope that it will not be misunderstood.

562-4 " However, inhibitors able to block endogenous signalling processes are often more valuable tools for investigation of physiological mechanisms than approaches that impose an overexpressed synthetic activity."

Both types of studies can have value, it depends on how they are deployed.

We agree with the reviewer that both approaches may have value. So we have replaced the words "are often " to "can be" (line 352, bold), as we think it is fair to say that overexpression is more prone to problems for physiological studies than loss of function approaches. Thus knockout mice are favoured for investigating physiological gene function over transgenics i.e. overexpressors, which are more commonly used as disease models.

Figure issues and suggestions

Figure 1

- this is at best a supplemental figure. It does not contain any newly acquired data presented here.
- epsilon should be defined in the figure legend and associated with a particular fluorophore
- emission and excitation spectra should be distinctly indicated (e.g solid vs dash lines)
- shade overlap region between FP and LOV2
- the -> marker should be replaced with an arrow
- 1C, LOV2 is labeled norm, 1G excitation is misspelled, 1E Lov2 absorption is used instead of excitation
- perhaps align spectra in which LOV2 is the emitter

We have moved this entire figure to supplementary data with the recommended changes. For the alignment, we arrange the figure so that all cases where LOV2 is considered a candidate donor as a single column.

We retained 1C-D y-axis labelling as absorption because the data available is an absorption spectrum not an excitation spectrum (Gauden et al., 2004).

Figure 2

- The images in fig. 2a are too small and blurry -
- figure Ei and Fi - the co-expression of the probes - are superfluous and add unnecessary complications
- the axes are labeled in an overly complex manner. Use the figure legends to explain the details.
- the boxes around each graph are unnecessary

We have reorganized Fig. 2. Fig 2A (now Fig. 1A) is allocated more space and should now be clearer. Figs. 2Ei and Fi were removed to supplementary data (explained above), we have removed the boxes and simplified the axis labels.

Figure 3

- the wavelength of light can be stated in the legend. The relevant parameter on the x-axis is the number of flashes

We have moved the wavelength to the legend.

Figure 4

- The reading number in figure 4 is a poor choice for the x-axis. the key aspect of these experiments is that there is an increasing delay between trials of each 20 readings. The x-axis should include this feature, by adding gaps of increasing size between each set of measurements. Alternatively, simply put 4C,D as supplementary material and focus the reader on 4E (4D and 4E are essentially redundant)
- Also 4B and C should be on the same graph. If they are on different graphs, they should be the same width at a minimum.
- 4D fluorescence is spelled incorrectly
- the boxes around each graph are unnecessary

We generated the modified chart as the reviewer suggested (new Fig. 3B). This represents more linearly how the experiment was carried out and helps the reader understand the timing involved. However, we think this is a relatively inefficient way to present and visualize subsequent results - the biggest changes are right at the start of the timecourse, and the linear representation is particularly crowded in this region so the curve of dequenching is hardly visible even though the graph occupies much more space. For this reason, we first show the linear representation followed by the earlier representation (now both constructs combined to one graph, as suggested). This way the reader can see how the original data was captured over time in practice, and then understand our non-linear projection of the data subsequently used in the rest of the manuscript. The latter spreads the early timepoints for easier visualization of the kinetics in each case. We have removed previous Fig 4D, we agree it was redundant, and there are no more boxes around the graphs.

Figure 5

- The reading number in figure 5 is a poor choice for the x-axis. the key aspect of these experiments is that there is an increasing delay between trials of each 20 readings. The x-axis should include this feature, by adding gaps of increasing size between each set of measurements.

As explained for former fig. 4 above, we have presented the linear representation and how the non-linear projection facilitates visualization of the changes in the new Fig 3. Now the non-linear projection has been explained and demonstrated to present the changes more clearly, we keep this for Fig. 4 (former Fig 5).

Figure 6

The results in figure 6 A-C are mostly redundant with the table.

We moved the table (old Fig. 4E) to the supplementary data because there is overlap between parameters in the second table (though calculated a different way). We believe the approach used for the latter (old Fig. 4I) is likely to be more accurate, more complete and in most cases more useful. All tables were removed from figures as per journal style.

Reviewers' Comments:

Reviewer #1:

Remarks to the Author:

-

Reviewer #2:

Remarks to the Author:

The revised manuscript is greatly improved, though it is not completely ready for publication at this stage, in part due to new information that appeared during the revision. The manuscript remains quite dense and jargon-rich.

1) The one control that would be important to add, for example in the case of mTq2 is to compare mTq2-optoNES with a version of mTq2-optoNES in which mTq2 is mutationally inactivated with a point mutation in the fluorophore. The authors show in figure 4 that "the N-terminal linker between tag and LOV2 also has a considerable impact" on the relaxation rate". Given that the sensitivity of an optogenetic probe is impacted by the relaxation rate (fig 8), the authors should directly demonstrate that the RET and not a non-specific aspect of the linker or fluorophore (as opposed to the ability to exchange energy) is responsible. This would have been a cleaner experiment than the increase in linker length. Specifically, the authors could measure dose response of nuclear export to light of two constructs that only differ in the presence of a functional fluorophore.

The authors results with V416I and I427V support their argument, but including that in the beginning of the manuscript would involve a large scale reorganization of the manuscript and it might create additional complexity in the manuscript.

2) The response to reviewer 1 point 1 is not satisfactory. While there are very specific cases where a 7-fold improvement in sensitivity can be achieved, those cases are quite limited therefore it need not be mentioned 7 times (lines 27, 56, 108, 115, 122, 364, and 439). The increased sensitivity is secondary to the ability to monitor switching in vivo and the authors are emphasizing the secondary point.

3) Line 23ff ""Yet cellular optogenetics applications remain limited in practice. Validation difficulties constrain most usage to easily verified protein translocation rather than inhibition of diffusible targets."

Suggested revision:

Cellular optogenetics applications remain limited with diffusible targets as the response of the probe is difficult to independently validate.

4) Line 30 and 63, if the increase in relaxation rates are mentioned in the abstract and introduction, then the authors should mention that this is a consequence of flanking sequences (and not a difference e.g. between in vivo/in vitro).

5) Line 44ff A second, perhaps greater impediment to cellular optogenetics is that direct in situ detection of optogenetic switching has not been achieved. Except in specific cases where protein translocation or other visualised downstream phenomena are targeted, it is hard to know whether an actuator has been switched.

This is a big exception and the first sentence should be rephrased.

6) Line 103ff, there is no good reason to include the results in Supplementary Figures 6,7, which are derived from a poorly designed experiment.

We thank the reviewers for their additional input to help us further improve the manuscript. Our point-by-point responses, detailing how the issues raised have been addressed and those changes we have made, are below. All changes are shown in the text in revision marks. These were made in response to the reviewers' input (as detailed below, text in lines referred to are shown in the text in bold), text related to the new experiment and removal of unnecessary words to make space in the word count for the new text. There are also as minor corrections in the methods, discussion (brightness of LOV2), and labelling of Figs. 1E/F, 9 and supp. fig. 1.

Reviewer #2 (Remarks to the Author):

The revised manuscript is greatly improved, though it is not completely ready for publication at this stage, in part due to new information that appeared during the revision. The manuscript remains quite dense and jargon-rich.

1) The one control that would be important to add, for example in the case of mTq2 is to compare mTq2-optoNES with a version of mTq2-optoNES in which mTq2 is mutationally inactivated with a point mutation in the fluorophore. The authors show in figure 4 that "the N-terminal linker between tag and LOV2 also has a considerable impact" on the relaxation rate". Given that the sensitivity of an optogenetic probe is impacted by the relaxation rate (fig 8), the authors should directly demonstrate that the RET and not a non-specific aspect of the linker or fluorophore (as opposed to the ability to exchange energy) is responsible. This would have been a cleaner experiment than the increase in linker length. Specifically, the authors could measure dose response of nuclear export to light of two constructs that only differ in the presence of a functional fluorophore.

The authors results with V416I and I427V support their argument, but including that in the beginning of the manuscript would involve a large scale reorganization of the manuscript and it might create additional complexity in the manuscript.

The reviewer points out that our data in Fig. 4 shows that sequence N-terminal to the LOV2 domain influences relaxation time, and therefore the elimination of FP-dependent response sensitivity caused by introducing a spacer N-terminal to LOV2 in Fig. 1 could be due to changes in relaxation time and not RET.

We agree this is an important issue and should be addressed. In Fig. 2-3 we already measured the relaxation times of the mTq2 constructs with and without spacer and found them to be identical (Fig. 2H, 3D; listed in table 1, along with the corresponding values for the Ypet and mScarlet constructs). Therefore we already included the required evidence that addresses this concern in Fig. 2, well before the sequence-dependence issue deriving from Fig. 4 arises. For this reason we think that the data in the previous version fully addresses the concern of the reviewer without any reorganisation of the manuscript that the reviewer hinted could be required.

We have now added a sentence in the results to explain that this potential issue is addressed (line #s 215-217, in bold).

Nevertheless, we wished to consider the suggestion of the reviewer to provide further experimental evidence on this point. The reviewer's proposal to use a non-fluorescent mTq2 would clearly prevent us from recording the dynamic changes in localisation in response to different illumination conditions and is not feasible. Addition of secondary fluorescent tags increases potential RET

pathways and likely complicates the situation. Therefore we evaluated a panel of different switchable proteins to acquire this additional evidence.

First we considered PS-CFP2 which switches from cyan-like to yellow-like properties, and PA-GFP switching from sapphire-like to yellow-like properties. However the large amounts of energy required make these tools suited to switching protein populations in subcellular regions - we did not have sufficient power to switch them entirely even in a single field whereas our experiments requires use of multiple fields. Moreover, the low-% switching we did observe was accompanied by evidence of photo-damage to the cells, and irreversible switching of these proteins would prevent comparison between different photon doses on the same cells, which was the basis for our assays to achieve the required sensitivity. Unfortunately there are no suitable reversibly switchable cyan proteins (we already showed in supplementary data that RET from TFP is limited).

Of the proteins we considered, Dronpa provided the best properties for the present purpose –

- i) reversible and complete switching to a dark 400nm-absorbing state (<10% residual fluorescence) by illumination with moderately intense 500nm light (0.2mol.m^{-2});
- ii) reversible switching back to a Ypet-like state with lower level 405nm illumination (2mmol.m^{-2}),
- iii) long thermal relaxation time, the recovery of fluorescent state from the dark state (reported in the literature as ~840 minutes, PMID: 18724362), ensuring that Dronpa switching does not occur spontaneously during the experiments;
- iv) relative insensitivity to 470nm light that can be used to switch LOV2. The 438nm light used to activate LOV2 in other experiments cannot be used as it also switches Dronpa. LOV2 absorbance at 438nm and 470nm are similar;
- v) After activation of LOV2 by 470nm light (using the protocol of Fig. 1, i.e. 10 flashes of activation light at 0.1Hz at different photon fluxes), the nucleocytoplasmic location of Dronpa could be observed from a single image at low level 500nm excitation (0.1s , $77\mu\text{mol.m}^{-2}$) immediately preceded with a 405nm flash (0.5s , 2mmol.m^{-2}). Thus even dark Dronpa acquires its fluorescent state without having time to translocate (which takes tens of seconds). For consistency, the 405nm-500nm acquisition protocol was used, regardless whether Dronpa-LOV2 was in dark state or already in fluorescent state (Supp. Fig. 9D).

We were able to repeatedly and reproducibly switch Dronpa from dark state to yellow-like state (Supp. Fig. 9E). After each switch, 5 minutes was allowed for LOV2 to relax and resting nucleocytoplasmic location to recover. A resting state 500nm image confirmed this recovery and was used as nucleocytoplasmic ratio baseline.

Using this approach, we found that the translocation of Dronpa-optoNES was more sensitive to photons when Dronpa was previously switched to a dark state than when it was in yellow fluorescent protein-like state (Fig. 1I, Supp. Fig 9F).

This provides an additional line of supporting evidence that the fluorescent properties of the attached tag influences the photon sensitivity of LOV2 switching and further strengthens our conclusion that RET is responsible for the changes in optoNES sensitivity in Fig. 1

We have included this new data as a bar-chart (Fig. 1I) corresponding to Figs. 1G-H, with the detailed schemes and response curves in supplementary data (Supp. Fig. 9). It is referred to in the results (lines 117-123). This new experiment has important additional implications. It demonstrates that we can not only engineer LOV2 sensitivity by RET at the design stage but we can also dynamically modulate sensitivity (by 405nm light and Dronpa state switching in this case) after delivery to cells. This is relevant also to the previous comment of referee 1 about whether bidirectional modulation of LOV2 sensitivity has any value. The ability to dynamically and reversibly change sensitivity of an individual LOV2 construct *in situ* potentially opens the door to the prospect of optogenetic

multiplexing, whereby orthogonal inputs (e.g. non-blue light) could be used to select which actuator species is most responsive to input (blue light). However, this is a distinct aim from that of the present study and considerable optimisation work remains before such a future goal could be achieved. Therefore we simply mention this future prospect when discussing the implications of the results (lines 366-368) and also mention in the introduction that dynamic modulation of sensitivity is possible (line 26).

2) The response to reviewer 1 point 1 is not satisfactory. While there are very specific cases where a 7-fold improvement in sensitivity can be achieved, those cases are quite limited therefore it need not be mentioned 7 times (lines 27, 56, 108, 115, 122, 364, and 439). The increased sensitivity is secondary to the ability to monitor switching in vivo and the authors are emphasizing the secondary point.

As specified in response to point 1 we envisage an additional use of suppression of LOV2 sensitivity. However, we have now limited the specification of a “7-fold” sensitization to a single mention in the results and once at the end of the discussion, to address the reviewer’s concern. We have removed the mention of 7-fold entries from lines 27, 56, 108, (line 115 was not about sensitization, explained below) and 364 (former line numbering).

Please note that the mention of “7-fold” in line 115 was a calculation of $[R/R0]^6$ for the estimated dipole-dipole distance change from 5 to 6.9nm caused by introduction of spacer. It was not a reiteration of sensitivity increase. This estimate was requested by reviewer 1, so we have retained it.

3) Line 23ff ""Yet cellular optogenetics applications remain limited in practice. Validation difficulties constrain most usage to easily verified protein translocation rather than inhibition of diffusible targets."''

Suggested revision:

Cellular optogenetics applications remain limited with diffusible targets as the response of the probe is difficult to independently validate.

We have made this change exactly as requested (line 23-24).

4) Line 30 and 63, if the increase in relaxation rates are mentioned in the abstract and introduction, then the authors should mention that this is a consequence of flanking sequences (and not a difference e.g. between in vivo/in vitro).

We have included this point in the abstract (line 29) and introduction (lines 64-65).

5) Line 44ff A second, perhaps greater impediment to cellular optogenetics is that direct in situ detection of optogenetic switching has not been achieved. Except in specific cases where protein translocation or other visualised downstream phenomena are targeted, it is hard to know whether an actuator has been switched.

This is a big exception and the first sentence should be rephrased.

We have rephrased this sentence as below (lines 44-46):

“A second difficulty optimizing optogenetic regulators for new targets is that real-time in situ detection of photosensor switching has not been reported.”

6) Line 103ff, there is no good reason to include the results in Supplementary Figures 6,7, which are derived from a poorly designed experiment.

We have removed the corresponding figures.

Reviewers' Comments:

Reviewer #2:

Remarks to the Author:

This revised manuscript has addressed all the remaining issues. I commend the authors on their thorough response to the reviewers points, which has resulted in an interesting, clear manuscript which will be useful to the optogenetic community.